



# Coarse and Giant Particles are Ubiquitous in Saharan Dust Export Regions and are Radiatively Significant over the Sahara

Claire L. Ryder[1], Eleanor J. Highwood[1], Adrian Walser[2], Petra Seibert[3], Anne Philipp[2], Bernadett Weinzierl[2]

[1]Department of Meteorology, University of Reading, Whiteknights, Reading, RG6 6BB, UK.
[2]University of Vienna, Faculty of Physics, Aerosol Physics and Environmental Physics, Vienna, Austria
[3]University of Natural Resources and Life Sciences, Institute of Meteorology, Vienna, Austria

*Correspondence to*: Claire L. Ryder (c.l.ryder@reading.ac.uk)

**Abstract.** Mineral dust is an important component of the climate system, interacting with radiation, clouds and biogeochemical systems, and impacting atmospheric circulation, air quality, aviation and solar energy generation. These impacts are sensitive to dust particle size distribution (PSD), yet models struggle or even fail to represent coarse (diameter (d) >2.5 µm) and giant (d>20 µm) dust particles and the evolution of the PSD with transport. Here we examine three state-of-the-art airborne observational datasets, all of which measured the full size range of dust (d=0.1 to >100 µm) at different stages during transport, with consistent instrumentation. We quantify the presence and evolution of coarse and giant particles and their contribution to optical properties. Observations are taken from the Fennec fieldwork over the Sahara and in the Saharan Air Layer (SAL) near the Canary Islands, and from the AER-D fieldwork in the vicinity of the Cape Verde Islands in the SAL.

Observations show significantly more abundant coarse and giant dust particles over the Sahara compared to the SAL: effective diameters of up to 20 µm were observed over the Sahara, compared to 4 µm in the SAL. Mass profiles show that over the Sahara 40% of dust mass was found in the giant mode, contrasting to 2 to 12% in the SAL. Size-resolved optical property calculations show that in the shortwave (longwave) spectrum excluding the giant mode omits 18% (26%) of extinction over the Sahara, compared to 1-4% (2-6%) in the SAL. Excluding giant particles results in significant underestimation of both shortwave and longwave extinction over the Sahara, as well as of mass concentration, while the effects in the SAL are smaller but non-negligible. Omitting the giant mode results in a greater omission of dust longwave radiative effects compared to the shortwave, suggesting a bias towards a radiative cooling effect of dust when the giant mode is excluded and/or the coarse mode is underestimated. This will be important in dust models, which typically exclude giant particles and underestimate coarse mode concentrations.

A compilation of effective diameters against dust age since uplift time suggests that two regimes of dust transport exist. During the initial 1.5 days, both coarse and giant particles are rapidly deposited. During the subsequent 1.5 to 10 days, PSD barely changes with transport, and the coarse mode is retained to a much greater degree than expected from estimates of gravitational



sedimentation alone. The reasons for this are unclear, and warrant further investigation in order to improve dust transport schemes, and the associated radiative effects of coarse and giant particles in models.

## 1. Introduction

Mineral dust aerosol is an important component of the climate system. Around 1,100 Tg yr$^{-1}$ of dust is uplifted annually, with around 57% of this originating from North Africa (Huneeus et al., 2011). Atmospheric mineral dust is estimated to account for 70% of the global aerosol mass burden and 25% of the global aerosol optical depth (AOD) (Kinne et al., 2006). During atmospheric transport and through subsequent deposition, dust exerts an impact on the climate system by interacting with both shortwave and longwave radiation (Tegen and Lacis, 1996; Liao and Seinfeld, 1998). These radiative effects can impact on the global energy balance, land and sea surface temperatures, atmospheric heating, and thus circulation patterns. Impacts can be particularly strong regionally where dust loadings are high, such as the Sahara desert, where dust affects North African atmospheric dynamics such as the Saharan heat low, Sahelian precipitation and North Atlantic hurricane development (e.g. Colarco et al. (2014); Pan et al. (2018); Lavaysse et al. (2011); Strong et al. (2018)). Additionally, dust particles can impact cloud development by acting as cloud condensation nuclei and ice nuclei (Kumar et al., 2011; Hoose and Mohler, 2012). Dust can affect atmospheric chemistry by providing a surface for heterogeneous reactions (Bauer et al., 2004). Dust is deposited to the oceans and Amazon rainforest providing nutrients to a variety of ecosystems (Jickells et al., 2005). Finally, dust is a natural hazard, having a negative impact on aviation and transport (Weinzierl et al., 2012), solar energy generation and air quality, and hence human health (Middleton et al., 2018). The annual economic cost of dust storms may reach into the billions of US dollars for certain countries (Middleton, 2017).

All of these impacts are sensitive to dust particle size (Mahowald et al., 2014). For example, dust size distribution can affect cloud interactions since smaller dust particles can be more hygroscopic (Ibrahim et al., 2018), while on the other hand larger particles can be more effective cloud condensation nuclei (Petters and Kreidenweis, 2007). Size distribution also affects surface area and therefore ice nucleation (Diehl et al., 2014). Larger particles contribute more to dust mass, which controls the impact of dust on ocean and tropical rainforest ecosystems (Jickells et al., 2005; Yu et al., 2015). A higher proportion of fine particles will lead to elevated PM2.5, and subsequent impacts on respiratory health (Middleton, 2017).

Dust size distribution has a strong impact on its radiative interactions (Tegen and Lacis, 1996). In the shortwave spectrum, a larger coarse mode reduces the single scattering albedo (SSA) of dust, causing more absorption of solar radiation and atmospheric heating. For example, Ryder et al. (2013b) found that including the coarse and giant modes over the Sahara resulted in the SSA dropping from 0.92 to 0.80 with an associated increase in atmospheric heating by up to a factor of 3. In the longwave spectrum, larger particles are able to exert a stronger radiative effect. For example, Otto et al. (2011) show that that including particles larger than 5 μm more than doubles the longwave aerosol optical depth (AOD). Together these radiative



effects can change the sign of the net radiative effect of dust and the impact of dust on atmospheric circulation (Woodage and Woodward, 2014; Strong et al., 2018). Additionally, satellite retrievals are sensitive to the assumed dust size distribution. Given these impacts of dust size distribution on climate and particularly radiation, it is important to have the best possible observations of dust particle size distribution (PSD) across all sizes, to understand its vertical distribution through the atmosphere, and how these change with transport.

Airborne observations are an important tool for probing the vertical distribution of dust size and concentration. Historically, optical measurement techniques have frequently been utilized - which require a conversion of scattered signal to particle size, and therefore incorporate uncertainties due to particle refractive index, shape and non-monotonic Mie scattering (Ryder et al., 2015; Ryder et al., 2013b; Walser et al., 2017). Many earlier measurements of dust were also limited by the maximum size measured (often not more than 10 µm diameter) or by sampling behind inlets which restricted the maximum particle size and passing efficiency (e.g. Ryder et al. (2018) Table 1). In the last ten years, airborne observations of dust have progressed to measuring significantly larger particle sizes, often on wing probes which do not suffer from inlet loss effects (Weinzierl et al., 2009; Ryder et al., 2013b). More recently, light shadowing measurement techniques, which do not require a scattering to size conversion, have been applied to particles larger than 10 µm diameter (Ryder et al., 2013b; Ryder et al., 2018). Finally, airborne observations have taken place in more remote Saharan desert regions, where larger dust particles are more likely to be prevalent (Ryder et al., 2015; Weinzierl et al., 2009).

As a result of these developments, observational campaigns have now shown that coarse and giant dust particles are far more prevalent, and transported further and higher than previously thought. Fennec, the Saharan Mineral Dust Experiment 1 (SAMUM1), Saharan Mineral Dust Experiment 2 (SAMUM2), Saharan Aerosol Long-range Transport and Aerosol-Cloud-Interaction Experiment (SALTRACE), AERosol Properties – Dust (AER-D) and Aerosol Direct Radiative Impact on the regional climate in the MEDiterranean region (ADRIMED) have all reported a significant presence of coarse to giant dust particles, despite the sampling locations of Saharan dust ranging from very close to sources to thousands of kilometres away (Weinzierl et al., 2009; Weinzierl et al., 2011; Weinzierl et al., 2017; Ryder et al., 2013b; Ryder et al., 2018; Denjean et al., 2016; Marenco et al., 2018).

Typically, dust models do not include particles larger than 20 µm diameter (Huneeus et al., 2011). Historically this has been because larger particles have been assumed to be rapidly deposited. However, recent work has shown that climate models face serious challenges in representing the dust cycle adequately, part of which stems from accurately representing dust PSDs. For example, Evan et al. (2014) find that CMIP5 climate models underestimate dust mass path (dust mass loading per square metre) by a factor of 3, 66% of which is due to a bias in size distribution skewed towards smaller particles. Kok et al. (2017) found that by using an observationally constrained dust emission PSD, global model calculations of dust radiative forcing were more positive (-0.48 to +0.20 Wm$^{-2}$) compared to previous estimates from AeroCom models (-0.6 to -0.3 Wm$^{-2}$) where smaller,





more cooling particles were over-represented and coarser, more warming particles were underestimated. As a result, observations of dust which include the coarse mode are in demand (Formenti et al., 2011b; Ansmann et al., 2011; Ansmann et al., 2017; Samset et al., 2018) for model validation. There are also implications for satellite optical models and retrievals since these also rely on accurate aerosol optical properties which are affected by PSD.

Here we contrast state-of-the art airborne observations of dust size at two stages representative of Saharan dust transport. We compare observations over the Sahara from the Fennec fieldwork to observations over the tropical Eastern Atlantic within the Saharan Air Layer (SAL), from both the AER-D and Fennec fieldwork. These observations fully include the coarse and giant modes of dust, measuring up to 100 µm for AER-D and 300 µm for Fennec. Both observational campaigns use consistent

instrumentation, utilizing wing probes and light shadowing techniques for the giant mode, thus evading some of the historical measurement challenges in dust observations. The Fennec dataset is particularly novel since it includes observations within 12 h of dust uplift in remote Saharan locations, where few other airborne measurements (if any) have been taken.

We contrast dust characteristics close to sources to those at the beginning of trans-Atlantic transport. We present mean size

distributions, vertical distributions of size metrics and vertical distribution of mass concentration for different size ranges, some of which data for Fennec has not previously been published. We then calculate optical properties as a function of size, using the ambient number concentrations measured, to illustrate the contribution of coarse and giant particles, using a range of the latest refractive indices from the literature. We include longwave scattering, which is frequently neglected. Finally, we put the Fennec and AER-D size distributions and dust age into context with published airborne observations to show the wider

context of transport of coarse and giant particles.

## 2.    Methods

In the literature the specific definition 'coarse' and 'giant' aerosol particles are not well defined. This is because the origins of aerosol mode size terminology relate to broad size modes, partly overlapping in size, relating to aerosol generation mechanism, composition and measurement technique (Whitby, 1978; Kulkarni et al., 2011). For example, the lower bound of the coarse

mode diameter has been defined as particles larger than the following: 1 µm (Lohmann et al., 2016; Mahowald et al., 2014), 2 µm (Kulkarni et al., 2011), 2.5 µm (often relating to PM2.5) (Neff et al., 2013; Seinfeld and Pandis, 2006; NASA, 2018), 5 µm (Kok et al., 2017), 10 µm (Renard et al., 2018). Similarly, giant particles are referred to as covering a wide size range upwards of 20 µm (Feingold et al., 1999), 37.5 µm (Ryder et al., 2013a), 40 µm (Jaenicke and Schutz, 1978), 62.5 µm (Goudie and Middleton, 2001) and 75 µm (Betzer et al., 1988; Stevenson et al., 2015). Weinzierl et al. (2011) do not define giant

particles, but start counting 'large coarse mode' dust particles upwards of 10 µm. Often the definition of coarse and giant particles are relative and case-study or instrument specific. In this paper we define the accumulation mode as 0.1<d<2.5 µm,





the coarse mode as d>2.5 µm and the giant mode as d>20 µm, since this is the diameter above which models rarely incorporate dust (Huneeus et al., 2011). Henceforth in this article, particle size is referred to in terms of diameter (d).

## 2.1. Size Distributions

This work exploits airborne observations taken during the Fennec project during June 2011 over both the Sahara desert and in the SAL in the vicinity of the Canary Islands (Washington et al., 2012; Ryder et al., 2015) and more recently over the Tropical Atlantic Ocean within the SAL during the AER-D project in August 2015 (Ryder et al., 2018). Figure 1 shows the location of the fieldwork. During both fieldwork projects, the FAAM BAe146 research aircraft was deployed, and size distributions of the full particle size distribution were measured by wing-probes (up to 300 µm during Fennec and up to 100 µm during AER-D), using a Passive Cavity Aerosol Spectrometer Probe, Cloud Droplet Probe and Cloud Imaging Probe-15 during Fennec and Passive Cavity Aerosol Spectrometer Probe, Cloud Droplet Probe and 2D Stereo Probe instruments during AER-D. Size distributions from both field campaigns have already been published: full descriptions of the instrumentation, uncertainties and findings are available for the Fennec observations over the Sahara (Fennec-Sahara: Ryder et al. (2013b)), the Fennec observations in the SAL (Fennec-SAL: Ryder et al. (2013a)) and the AER-D observations in the SAL between the Cape Verde and Canary Islands (AER-D SAL: Ryder et al. (2018)), as well as specific flight locations, tracks, and details of dust events sampled.

For Fennec-Sahara and AER-D-SAL, observations from horizontal flight legs are available (117 from Fennec-Sahara, 19 from AER-D-SAL), which capture some of the spatial variability in dust properties. Horizontal flight leg data are not available for Fennec-SAL, where only take-off and landing profile observations were made. For all three campaigns observations from aircraft profiles are available (21 from Fennec-Sahara, 31 from AER-D-SAL, 21 from Fennec-SAL), which capture a more complete altitude range. Fennec-Sahara profiles do not extend all the way to the surface due to aircraft operating restrictions. In addition, both the Fennec-Sahara horizontal flight legs and profiles are separated in to fresh, aged or uncategorized dust events (see Section 2.3).

Besides presenting the nature of the full size distributions, we calculate two size metrics representing the full PSD. These are maximum size detected ($d_{max}$) and effective diameter ($d_{eff}$) calculated directly from the aircraft-measured PSDs during horizontal flight legs. Effective diameter ($d_{eff}$) is a commonly used metric (Hansen and Travis, 1974), representing an area-weighted mean diameter. $d_{max}$ was initially used by Weinzierl et al. (2009) and is a useful indicator of transport of the largest sizes, which dominate the mass fraction. Here we use a simple estimation of $d_{max}$ as described in (Ryder et al., 2018), where $d_{max}$ represents the maximum particle size during a flight leg where at least 4 particles were detected within a single size bin. This implicitly represents the maximum size measured when concentrations of dust exceed $10^{-5}$ cm$^{-3}$ (or 10 m$^{-3}$) for a 20 minute flight segment for a particle size of 30 µm. Full details are provided in Ryder et al. (2018). This metric has not been previously published for the Fennec data. We also provide dust mass profiles calculated using the measured PSDs and assuming a density



of 2.65 gcm$^{-3}$ (Hess et al., 1998) which is representative of quartz particles (Woodward, 2001; Haywood et al., 2001; Kandler et al., 2009; Chen et al., 2011), taking data from aircraft profiles. Finally, we also calculate dust mass path (DMP) as in (Ryder et al., 2018): the vertically integrated mass of dust per unit surface area which has been used in satellite and model evaluations (Evan et al., 2014). All size distributions, size metrics and mass concentrations are provided at ambient conditions.

Here we expand on the existing published work and data from Fennec and AER-D. Our emphasis is on using the combination of data in the context of transport time and vertical distribution. We also provide some data from Fennec which was previously unpublished: vertical distributions of mass concentration, $d_{max}$, and separation of $d_{eff}$ between fresh and aged dust events, and the mean Fennec-SAL data.

We calculate optical properties utilizing the lognormal size distributions (since they are easily reproducible). For Fennec-Sahara and AER-D SAL, the lognormal PSDs are taken from horizontal flight legs, representing the range of observations encountered, as shown in Figure 2. For Fennec-Sahara, lognormal PSDs are provided in Ryder et al. (2013b). Here we use the mean logfit curves and as bounds of uncertainty on the PSD we also use the maximum and 10$^{th}$ percentile logfit curves (orange shading in Figure 2). The 10$^{th}$ percentile PSD (data given in supplement) is selected as the lower bound since the minimum curve for Fennec-Sahara presented in Ryder et al. (2013b) is an outlier of one case with extremely low dust loadings. For AER-D-SAL, we use the mean logfit curve, bounded by the minimum and maximum given in Ryder et al. (2018). For Fennec-SAL, only profile data is available (not horizontal flight legs). Therefore a logfit curve is fitted to the mean observational profile data from Ryder et al. (2013a) as shown by the blue line in Figure 2 (data available in supplement). The spread of PSDs for Fennec-SAL (blue shading) is narrower compared to the other two PSDs because the minimum and maximum represent the standard error of the mean as given in Ryder et al. (2013a).

### 2.2. Optical Property Calculations

In order to calculate dust optical properties, the Fennec and AER-D mean size distributions (Section 2.1) are used in combination with a range of literature refractive index data and a Mie scattering code, implying a spherical assumption. Although observations show that dust is not spherical, here we retain this simplification in order to allow a range of fast calculations, and also because many climate models assume spherical properties. In the longwave spectrum, non-sphericity effects of dust are not significant (Yang et al., 2007). Kok et al. (2017) show that dust non-sphericity increases shortwave extinction efficiency by around 50% for coarse particles, so therefore our results represent a lower bound on the impact of the coarse mode in the solar spectrum.

Spectral refractive index (RI) data, where the real part represents scattering and the imaginary part represents absorption, are taken from a range of sources. For the full spectrum, RI data are available from the OPAC database (Hess et al. (1998), based on values from d'Almeida et al. (1991) and Shettle and Fenn (1979)),  Volz (1973), Balkanski et al. (2007) assuming a 1.5%





hematite content, the World Meteorological Organization (WMO, 1983) and Fouquart et al. (1987). For the shortwave spectrum RI data are also available from Colarco et al. (2014) and for the longwave spectrum data are available from Di Biagio et al. (2017), where we have selected the Mauritania subset as it is representative of being middle-of-the range for their North Africa samples. Values are shown in Figure 3. At 0.55 µm these datasets yield real values of 1.52-1.53 and imaginary

components of 0.0015 to 0.0080. The Balkanski et al. (2007) and Colarco et al. (2014) datasets represent significantly more recent estimates of refractive index: Balkanski et al. (2007) estimate refractive indices assuming a central (1.5%) content of hematite when hematite is embedded in a matrix of clay and RIs are calculated assuming a dielectric mixture. Colarco et al. (2014) combine refractive indices from Colarco et al. (2002) from Total Ozone Mapping Spectrometer satellite retrievals at ultraviolet wavelengths and Kim et al. (2011) from AERosol Robotic NETwork (AERONET) at visible wavelengths. Both of

the latter two produce significantly lower imaginary parts, 0.0015 and 0.0024 at 0.55 µm respectively, widely considered to be more appropriate for accurately representing dust properties and consistent with recent observations (Rocha-Lima et al., 2018). In the longwave spectrum there is more variability between the RI datasets compared to the shortwave. We highlight the use of the much more recent and higher spectral resolution Di Biagio et al. (2017) dataset. The older (pre-2000) longwave datasets were limited in applicability due to being collected at limited geographic locations, being based on unknown mineral

composition, they may have been subject to unknown physio-chemical ageing and only Fouquart et al. (1987) satisfies the Kramers-Kronig relationship (Di Biagio et al., 2017).

In order to illustrate the impact of coarse particles on dust optical properties, firstly we calculate optical properties for the three mean PSDs, and also their uncertainties which are calculated from the shaded PSD range shown in Figure 2 for each campaign,

which represent the variability in the PSD, and also each of the refractive index datasets described above. Secondly, optical properties are calculated with a gradually incrementing maximum cut-off diameter for each PSD, in order to show how the optical properties depend on the maximum size considered, and how this differs for the three different PSDs measured during Fennec and AER-D. This enables the contribution of coarse and giant particles to the optical properties to be quantified. For these calculations only two wavelengths are selected: 0.55 and 10.8 µm. 0.55 µm since it represents the peak intensity of the

solar radiation spectrum, and 10.8 µm since extinction from dust at this wavelength is typically quite high, it falls within the atmospheric window where dust is able to exert a strong radiative effect, it avoids ozone and water vapour absorption channels, and it is also representative of one of the Spinning Enhanced Visible and Infrared Imager (SEVIRI) dust red-green-blue (RGB) channels (Brindley et al., 2012). Different thermal infrared wavelengths were also tested, and sensitivity to chosen wavelength in the results in Section 3.2.2 was found to be low.

**2.3.  Estimation of Dust Age**

Estimates of dust age for Fennec-Sahara and AER-D since uplift are taken from Ryder et al. (2013b) and Ryder et al. (2018) respectively. Briefly, for both campaigns, broad geographic dust source locations have been identified using SEVIRI dust RGB thermal infrared satellite imagery product (Lensky and Rosenfeld, 2008). Dust events sampled by the aircraft are tracked



backwards in time visually which allows determination of dust uplift time and location, and therefore dust age. For Fennec, this technique was combined with back trajectory analysis from Hybrid Single-Particle Lagrangian Integrated Trajectory model (HYSPLIT) (Draxler and Hess, 1998) and from FLEXible PARTicle dispersion model (FLEXPART) (Stohl et al., 2005). For AER-D, every dust event sampled could be linked to a haboob originating from a mesoscale convective system. For AER-D,

only SEVIRI imagery was used for dust source identification since for each case HYSPLIT back trajectories indicated different dust source locations, likely due to poor meteorological representation over the Sahara when convection was important (Ryder et al., 2018). Dust ages for Fennec-SAL are not included here since their values have been found to cover an extremely large range of times (Ryder et al., 2013a).

As in Ryder et al. (2013a,b), Fennec-Sahara data are also separated into 'fresh' and 'aged' categories, where fresh represents dust sampled in under 12 h since uplift time. Of the 119 sampling legs performed, 22 were fresh, 55 aged, and 40 uncategorized. Of the 21 Fennec-Sahara profiles, 5 were fresh and 16 aged.

The age of two SALTRACE dust samples from Weinzierl et al. (2017) measured over the western and eastern Atlantic were
derived from new backward simulations with the Lagrangian particle dispersion model FLEXPART (Stohl et al., 1998; Seibert and Frank, 2004; Stohl et al., 2005), using meteorological fields from the European Centre for Medium Range Weather Forecasts' ERA5 reanalysis (0.25°, 1 h resolution) as input. A generic aerosol species with a mean mass diameter of 7.9 μm and logarithmic standard deviation of 2.5 was tracked back from the five selected flight segments in each location, including the effects of gravitational settling, dry and wet deposition. The model produced source-receptor sensitivity values for a 50 m
layer adjacent to the ground. These sensitivities were multiplied with gridded, time-dependent dust emissions from the Copernicus Atmosphere Monitoring Service global natural emissions data set to obtain the corresponding contribution to the mass. The sum of the contributions over all grid cells at each of the time steps produced thus the simulated age distribution of the sampled dust aerosol. For both the eastern and western observations, the flight legs have been separated into five segments and ages calculated separately for each. The best-estimate of the SALTRACE dust age is given by the median for the segment
with the highest receptor mass concentration, while the uncertainties are given by the minimum and maximum 25th and 75th percentile ages across all five segments.

## 3. Results

### 3.1. Size Distributions, Mass Concentration and Vertical Distribution

The mean logfit volume size distributions from Fennec and AER-D and their variability is shown in Figure 2. Overall Figure
2 shows the following features which will be important in terms of optical properties: a strong giant mode for Fennec-Sahara and subsequent loss of this by Fennec-SAL and AER-D SAL; an enhanced accumulation and coarse mode for AER-D SAL relative to Fennec-Sahara and Fennec-SAL.



As expected, over the Sahara the giant mode (d>20 µm) is enhanced compared to the SAL. The Fennec-Sahara PSD peaks at 20-30 µm, while the AER-D-SAL PSD peaks at ~5 µm and the Fennec-SAL PSD peaks at 10-12 µm. In these cases, this can be explained by a greater dust age and distance from dust sources contributing to loss of the giant mode.

The accumulation and coarse mode are enhanced in AER-D-SAL compared to Fennec-Sahara and Fennec-SAL, with higher concentrations below 10 µm. However we did not observe this enhancement when the same dust events were observed in Fennec-Sahara and Fennec-SAL, rather the accumulation and coarse modes decreased in concentration from Fennec-Sahara to Fennec-SAL. The AER-D-SAL accumulation and coarse mode enhancement may occur because AER-D simply sampled more intense dust events, though this seems unlikely given that the Fennec dust events were also often very intense and AODs were mostly higher than AER-D (Ryder et al., 2015). This enhancement of the accumulation mode is similar to differences between SAMUM1 (Morocco) and SAMUM2 (Cape Verde region), where enhancements in number concentration between 0.3 to 4 µm during SAMUM2 were assigned to coagulational growth (Weinzierl et al., 2011). A number of the AER-D data segments were collected further south, closer to the intertropical convergence zone in moister conditions. Therefore another possibility is that hygroscopic growth took place, although generally dust is considered unlikely to react hygroscopically in this way (Denjean et al., 2015). Satellite imagery indicated clouds developed in the vicinity of every dust event sampled during AER-D-SAL during transport over the Sahara. Therefore, there is a possibility that the dust was affected by cloud or water vapour recycling during its transport journey, which may have allowed some form of coagulation, potentially impacting the size distribution (Ryder et al., 2015; Diaz-Hernandez and Sanchez-Navas, 2016; Weinzierl et al., 2011). Another possibility is that a slight difference in the dust sources activated between Fennec and AER-D led to different size distributions being mobilized initially.

Figure 4 demonstrates how dust size for Fennec-Sahara and AER-D-SAL change with altitude over the desert and in the SAL. AER-D datapoints at z<100m are marine boundary layer samples and are not discussed. Both $d_{eff}$ and $d_{max}$ show much larger values at all altitudes in Fennec-Sahara compared to AER-D-SAL. Over the Sahara $d_{eff}$ and $d_{max}$ drop off sharply with altitude while in the SAL they are more homogeneous in altitude. For Fennec-Sahara $d_{max}$ varied from 90 to 300 µm beneath 600 m while above 3.5 km $d_{max}$ varied from 15 to 180 µm. Contrastingly, values for AER-D-SAL were 20 to 80 µm. Particles sized over 20 µm (100 µm) were detected in 99% (89%) of the Fennec-Sahara dust layers. The impact of decreasing size with increased transport can also be seen in Figure 4b - AER-D-SAL $d_{eff}$ values are much lower than those for Fennec-Sahara, with a range of 3.6 to 4.0 µm in the SAL compared to 1.8 to 20.5 µm over the Sahara.

The largest $d_{eff}$ and $d_{max}$ values in Figure 4 are clearly dominated by fresh dust events (under 12 h since uplift). However, even for aged dust events (over 12 h since uplift, circles) very large particles were encountered, including at high altitudes: for Fennec-Sahara aged dust $d_{max}$ reached 195 µm beneath 1.5 km and 210 µm above 1.5 km, while $d_{eff}$ reached 10.7 µm beneath


1.5 km and 10.5 µm above 1.5 km. Aged $d_{eff}$ values over the Sahara are fairly homogeneous in the vertical. These large values at high altitudes indicate that the coarse and giant dust particles are entrained and transported in the atmosphere on longer than superficial timescales, and that for very fresh dust the coarse and giant mode are particularly enhanced at low altitudes.

Weinzierl et al. (2011) performed a similar comparison of $d_{max}$ between SAMUM1 and SAMUM2. Their results are not directly comparable to ours due to different instrumentation. However, relative altitude dependencies and changes during transport can still be compared. During SAMUM1, dust was well-mixed vertically, showing no altitude dependence of size and being similar to that of the aged dust from Fennec. Weinzierl et al (2011) also saw a decrease in $d_{max}$ between dust closer to sources in SAMUM1 (90% of cases had particles larger than 20 µm) to low altitude winter-time dust sampled over the Atlantic in
SAMUM2 (33% of cases had particles larger than 20 µm), similar to the $d_{max}$ decreases between Fennec-Sahara and AER-D-SAL.

Figure 5 shows the vertically resolved mass concentrations, since they are frequently used as a model diagnostic and biogeochemical cycles are also impacted by dust mass. Total mass concentrations (panel a) were notably higher at all altitudes
during Fennec-Sahara, gradually decreasing with altitude. In the SAL, mass concentrations were lower, and peaking in the SAL between 2 to 4 km for AER-D and being extremely homogeneous in height for Fennec-SAL upwards of 1 km. Fennec-Sahara mass concentrations can be extremely high, especially at lower altitudes, with the 75th percentile reaching values of up to 1940  µg m$^{-3}$. Contrastingly, the mass concentration in the accumulation mode (panel b) is highest during AER-D-SAL, which is a reflection of the enhanced accumulation mode shown in Figure 2. For Fennec-Sahara, there is a sharp increase in
the accumulation mode mass concentration beneath 1.4 km. Above 1.5 km, Fennec-SAL displays a similar profile to Fennec-Sahara, albeit in lower concentrations in keeping with the reduced concentrations shown in Figure 2. Given that the World Health Organization guidelines for air quality particulate matter limits for 24 hour mean PM2.5 and PM10 are 25 and 50 µg m$^{-3}$ respectively, the observations in Figure 5 are often well above these values, reinforcing the hazardous nature of dust events.

In Figure 5c and d the fraction of mass found at sizes greater than 5 and 20 µm diameter is shown. As in Ryder et al. (2018) these sizes are selected since they represent diameters at which models begin to underestimate the concentration of coarse particles (5 µm), and at which models have an upper limit (20 µm) (Kok et al., 2017). It is clear in panel c that during Fennec-Sahara the vast majority of dust mass was present at sizes greater than 5 µm (an average of 93% beneath 4.5 km), similar to Fennec-SAL (89% between 1 and 5 km) and also a large amount during AER-D-SAL (61% between 1 and 4 km in the SAL).
Since models begin to underestimate dust concentration at sizes above 5 µm diameter, a very large fraction of mass will be neglected. Similarly, during Fennec-Sahara, sizes greater than 20 µm diameter were still found to contain 40% of the dust mass beneath 4.5 km (panel d), or up to 68% for the 75th percentile. For AER-D-SAL and Fennec-SAL 2% and 12% of total mass respectively was found at these large diameters, though the 75th percentile reaches up to 19% and 56% respectively. Since 20 µm is typically the maximum diameter represented by dust models, a large fraction of dust mass over the Sahara is being

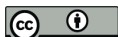



completely excluded from models, and although the percentage of mass found at sizes larger than 20 µm is fairly small on average, individual event values can reach much higher values, which will also be excluded by most models.

Mean DMPs are calculated at 3.2 gm$^{-2}$ (0.8 to 12.1 gm$^{-2}$) for Fennec-Sahara, 1.5 gm$^{-2}$ (0.2 to 6.2 gm$^{-2}$) for AER-D-SAL and
1.4 gm$^{-2}$ (0.2 to 2.3 gm$^{-2}$) for Fennec-SAL. As expected, mean values over the Sahara are higher compared to the SAL. All these values are much higher than those produced by models, such as the CMIP5 models analysed by Evan et al. (2014) with values of 0.05 to 0.46 gm$^{-2}$ with a multi-model median of 0.26 gm$^{-2}$ in the geographic region of the AER-D-SAL observations. Although the aircraft data only represent periods of around 3 weeks for each campaign, aerosol optical depths (AODs) were found to be climatological (Ryder et al., 2013b; Ryder et al., 2018), though they do represent the dustier summer months, while
the satellite and model data referred to here are annual means.

### 3.2.  Optical Properties

#### 3.2.1. Spectral Optical Properties

Figure 6 shows the spectral extinction coefficient calculated from the campaign-mean full PSDs shown in Figure 2 and the range of refractive index datasets described in Section 2.2. For clarity only Fennec-Sahara and AER-D-SAL are shown. In the
shortwave spectrum, it is clear that the size distribution difference between Fennec-Sahara and AER-D-SAL dominates the impact on extinction, with the AER-D-SAL PSD resulting in higher extinction due to the greater number concentration between 0.5 to 8 µm diameter in AER-D-SAL compared to Fennec-Sahara. As a result, Fennec-Sahara extinction is a factor of 0.7 less than AER-D-SAL (panel b). The extinction at these wavelengths is dominated by scattering (as opposed to absorption). As the RI real parts (relevant for scattering) are similar in all cases (even though the imaginary part varies) this causes little difference
to the total extinction, and therefore the size distribution is the dominant influence on extinction.

However, in the longwave spectrum, both PSD and RI are important. Different combinations of RI and PSD can give different spectral variation of extinction. Overall, the Fennec-Sahara PSD produces a higher extinction, by up to a maximum factor of 3.3 for the Di Biagio RI dataset. This is due to the increased scattering and absorption from the larger particles in the Fennec-
Sahara PSD. Interestingly, the application of the Fennec-Sahara PSD rather than the AER-D-SAL PSD is to dampen the spectral variability of extinction in the 7 to 12 µm spectral region: exactly the region utilized by satellite retrievals to detect dust. Thus, similar to Banks et al. (2018), we find that the coarsest dust may pose a challenge to longwave satellite detection algorithms by allowing coarse dust to effectively 'hide.'



### 3.2.2. Size-resolved Optical Properties

So far, we have shown how the different PSDs contribute to different spectral extinction properties. Here, we examine the size-resolved contribution to extinction coefficient at specific wavelengths (0.55 and 10.8 µm) in order to see how important the inclusion of a specific size range is to the optical properties.

Figure 7 shows the shortwave size-resolved percentage contribution to absorption (lightweight lines) and extinction (bold lines) coefficients at 0.55 µm for three different PSDs (different colours). In each case, the campaign mean PSD (as shown in Figure 2) and Colarco RI are used, as they represent central values. This is shown both as a percentage contribution to the total extinction (panel a), and cumulatively (panels b and c) to illustrate the cut-off diameter at which the majority of the extinction

is captured. In panels b and c, the shading represents the uncertainty to both the ranges of PSD shown in Figure 2 and the range of refractive indices tested.

For AER-D-SAL, Figure 7a shows that the main extinction contribution (thick black line) comes from particles sized around 1 µm and 3 µm. The scattering percentage contribution is not shown since it is very similar to the extinction curve since the

extinction is dominated by scattering. However, the absorption (thin black line) is dominated by a contribution from larger particles, with most absorption coming from particles sized around 5 µm. The Fennec-Sahara PSD (orange lines) shows an influence of much larger particles. In addition to the peaks at 0.9 and 3 µm, the largest extinction comes from 14 µm diameter particles. Similarly for absorption (thin orange line), the Fennec-Sahara optical properties are strongly dominated by the giant mode, with a peak contribution from 20 µm diameter particles. The properties of the Fennec-SAL dataset lie in between the

other two datasets, with peak contributions to extinction at 10 µm diameter and peak contribution to absorption at 12 µm diameter. The size-resolved extinction and absorption curves are a direct reflection of the shape and abundance of the different PSDs shown in Figure 2.

Figure 7b and c show the same results but in cumulative form, as well as the uncertainties around these curves due to the

variability in RI dataset and the range of PSDs sampled during the fieldwork. It can clearly be seen that the cumulative optical properties increase much more slowly as a function of diameter for Fennec-Sahara compared to AER-D-SAL and Fennec-SAL due to the effect of the greater concentration of giant particles in Fennec-Sahara. Only representing dust particles sized up to 20 µm diameter, as in many dust models, represents 99% (99-100%) of extinction in AER-D-SAL and 96% (96-97%) of extinction in Fennec-SAL, but only 82% (77-92%) of the extinction over the Sahara (Fennec-Sahara) (see also Table 1).

(Uncertainties are propagated from the range of PSDs and RI datasets). Besides the impacts on extinction, there are impacts on absorption: representing only up to 20 µm diameter results in 98% (97-100%) and 90% (87-91%) of absorption being represented for AER-D-SAL and Fennec-SAL respectively, but only 61% (52-82%) of absorption being represented for Fennec-Sahara. Whilst total extinction drives Aerosol Optical Depth, absorption drives shortwave atmospheric heating and



may subsequently impact regional circulation and the semi-direct effect. We note that these figures are lower bound estimates of the impact of neglected absorption and extinction in dust models, since they only account for giant particles being excluded, and not any underestimation of the coarse mode, which is included, but poorly represented in models (e.g. Kok et al. (2017); Evan et al. (2014)). It is also evident that by only representing sizes up to 2.5 μm, the majority of extinction is omitted (only

27, 48 and 31% of extinction for Fennec-Sahara, AER-D SAL and Fennec-SAL respectively is captured). This result emphasizes that it is crucial to measure the coarse mode of dust aerosol in order to fully capture its optical properties, and dust observations sampling only PM2.5 or behind size-restricted aircraft inlets will not provide a realistic representation of dust size and associated optical properties.

Figure 8 shows the size resolved contribution to optical properties but for a wavelength of 10.8 μm, representing the longwave spectrum. In Figure 8a, for AER-D-SAL and Fennec-SAL, the main contribution to extinction comes from particles sized around 6 μm and 10 μm diameter respectively, while the main contribution for Fennec-Sahara comes from particles sized 13 μm diameter. There is little difference in the relative contributions from scattering and absorption at this wavelength, with both contributing roughly equal amounts to the extinction (SSA values of 0.4-0.5). Figure 8b shows the same results cumulatively.

As with the results from the shortwave spectrum, much of the extinction for AER-D-SAL results from particles smaller than 10 μm diameter, while extinction for Fennec-SAL and Fennec-Sahara rises more slowly as a function of maximum diameter. Representing particles up to 20 μm diameter captures 98% (98-100%) and 94% (91-94%) of the extinction for AER-D-SAL and Fennec-SAL respectively, but only 74% (66-89%) for Fennec-Sahara (see also Table 2) – i.e. 26% (11-34%) of extinction at a wavelength of 10.8 μm is missed by not including any representation of giant dust particles over the Sahara. Also,

representing only up to 2.5 μm (such as done by PM2.5 observations or many observations behind aircraft inlets) results in only 2, 9 or 3% (for Fennec-Sahara, AER-D SAL and Fennec-SAL respectively) of the total extinction being captured.

Sensitivity to behaviour of the extinction curves at different wavelengths was tested, but no significant differences in the size-resolved behaviour was found, although the total extinction is different (as shown in Figure 6). The cumulative curves for

scattering and absorption at 10.8 μm are also very similar and are therefore not shown separately for the longwave (in contrast to the shortwave spectrum). This is consistent with Sicard et al. (2014) who showed that the effects of dust LW scattering are significant, and can cause up to a 50% underestimate at the TOA if neglected (Dufresne et al., 2002; Coelho, 2006).

### 3.3. The wider context of dust size and transport

Figure 9 compares the AER-D-SAL and Fennec PSDs to previous aircraft observations of Saharan dust from the last ten years

which fully observed the presence of the coarse and giant modes, at least up to 20 μm diameter: SAMUM1 (Weinzierl et al., 2009), SAMUM2 (Weinzierl et al., 2011), GERBILS (Johnson and Osborne, 2011) and ADRIMED (Denjean et al., 2016) and SALTRACE observations over the eastern and western Atlantic (Weinzierl et al., 2017). For the SALTRACE PSDs, the sub- and supermicron data shown in Weinzierl et al. (2017) have been combined and collectively inverted, guaranteeing a consistent





propagation of measurement uncertainties (in optical particle counter response, optical particle properties etc.) for the complete size range. Although other studies and fieldwork campaigns have also measured dust size distributions, here we focus on the coarse and giant modes and therefore only include studies which measured d>20 µm (and therefore do not include airborne observations from the DABEX, AMMA and NAMMA campaigns (Osborne et al., 2008; Chou et al., 2008; Formenti et al.,

2011a; Chen et al., 2011). Details of the instrumentation operated in each fieldwork campaign and relevant size limitations and maximum size measured are provided by (Ryder et al., 2018) in their Table 1. We do not extrapolate the PSD modes beyond the size measured (e.g. 20 µm for ADRIMED).

Overall, although the size distribution of dust shown in Figure 9 varies, it is clear that there is always a significant contribution

from dust particles sized d>5 µm, and when dust is closer to the source, there is also a strong contribution from particles larger than 20 µm diameter.

Clearly the size distribution of Saharan dust can be highly variable. However, the two campaigns measuring the greatest abundance of coarse and giant particles with d>10 µm were Fennec-Sahara and SAMUM1, both taking observations in remote

desert locations closer to dust sources. Volume mean diameters (VMDs) calculated from the mean PSDs (or envelope of PSDs for SAMUM) were also larger, at 21 µm for Fennec-Sahara and 5-14 µm for SAMUM1. AER-D-SAL, GERBILS, SAMUM2, Fennec-SAL and SALTRACE, further afield from dust sources, measured fewer giant particles, with maximum dV/dlogD at around 3 to 5 µm. Giant particles were present at 20-30 µm, but vastly reduced in volume concentration compared to Fennec-Sahara and SAMUM1. VMDs were lower at 3-4 µm (SAMUM2), 4 µm (GERBILS), 5.6 µm (AER-D-SAL), 12 µm (Fennec-

SAL) and 10-12 µm (SALTRACE E and W). These values represent the means of each campaign, and there will therefore be some additional overlap due to instrumental uncertainties and spatial and temporal variability within campaigns, though this data is not always available from the individual publications.

SAMUM2 represents dust transported over the Atlantic during winter at low altitudes. Although GERBILS observations were

made over the west African continent during summer, it is likely that the dust events sampled represented aged regional dust with a depleted coarse mode (Haywood et al., 2011; Johnson and Osborne, 2011). ADRIMED also represents transported dust, but over the Mediterranean Sea. At diameters of 20 µm ADRIMED volume concentrations are similar to AER-D-SAL and SAMUM2, with a suggestion of a very large giant mode at even larger diameters (e.g. Figures in Denjean et al. (2016)). AER-D-SAL also represents transported dust, and accordingly sits closer to GERBILS and SAMUM2 in Figure 9 than to Fennec-

Sahara and SAMUM1.

Figure 10 shows dust effective diameters as a function of estimated dust age since uplift. Firstly, Figure 10a shows Fennec-Sahara and AER-D-SAL separated by dust events. Fennec-SAL is excluded because the range of dust ages is too broad for it to be a useful addition (Ryder et al., 2013a). During AER-D-SAL, the estimated dust age varied from 0.7 to 4.6 days, while



the range of effective diameters was very small, with flight-means between 3.9 to 4.2 μm. Uncertainties in dust age for flights b928 and b934 are much larger due to the possibility of dust uplift from multiple sources along the transport pathway. Despite AER-D-SAL flights measuring dust with a range of transport times, the effective diameter showed only a variation of 5% about the mean of 4.0 μm. This contrasts sharply to observations of fresher dust from Fennec-Sahara where $d_{eff}$ showed a

decreasing trend with dust age. For Fennec-Sahara the freshest dust events (under 12 h since uplift) had mean $d_{eff}$ values of 8 to 13 μm, dropping to a mean of 6 μm for dust aged around 2 days. The addition of the data from AER-D-SAL suggests that in the bigger picture, dust size distributions change rapidly following initial uplift and transport, depositing some fraction of both coarse and giant particles, but after around 2 days size distribution appears to stabilize.

Figure 10b shows $d_{eff}$ against dust age since uplift for a range of airborne fieldwork campaigns, after Ryder et al. (2013a) (their Figure 11) and Denjean et al. (2016) (also their Figure 11). However, here we show $d_{eff}$ for the full size distribution (0.1 to 300 μm, or up to the maximum size measured in each campaign as shown in Figure 9), since dust particles are present in both the submicron sizes (Formenti et al., 2011b) and at d>20 μm (in contrast to Denjean et al. (2016), where $d_{eff}$ representing solely 1-20 μm was presented, and consequently their values are higher). GERBILS data yield a mean effective diameter of around

3 μm, but are not included in Figure 10b as no estimate of dust age was provided, though dust was likely to be relatively aged rather than fresh (pers. comm. B. Johnson). This analysis is different to previous compilations of dust size observations (e.g. Reid et al. (2008); Formenti et al. (2011b)) because we 1) relate dust size to time since uplift, 2) only include airborne observations (since elevated dust properties are often different to those at the surface), 3) only include observations which measured at least up to 20 μm diameter unencumbered by inlet restrictions, and 4) incorporate more recent data – particularly

that from Fennec which provides data from the remote Sahara very close to dust uplift time, and SALTRACE, providing tran-Atlantic observations.

Figure 10b shows that the stabilization of the size distribution indicated in Figure 10a still holds once other airborne data are included. Very large particles are evident immediately after uplift with high mean $d_{eff}$ values of 6 to 10 μm. $d_{eff}$ decreases

rapidly until around 1.5 days after uplift, after which the observations suggest little change in $d_{eff}$ from around 2 days' transport onwards.

The range of $d_{eff}$ values at over 1.5 days' transport in Figure 10b is fairly wide (from 1.4 to 5.2 μm). SAMUM2 data shows a slightly lower mean $d_{eff}$ value (2.4 μm) compared to AER-D-SAL, ADRIMED and SALTRACE (3.9 to 5.0 μm), though this

may be a result of SAMUM2 observations being taken in the winter season when dust is transported by different meteorological mechanisms and uplifted to lower altitudes over the Sahara (McConnell et al., 2008; Knippertz and Todd, 2012; Tsamalis et al., 2013), which may influence size distribution differences. Focusing solely on the summertime campaign data, the spread of $d_{eff}$ values is very narrow, even after 9 days' transport across the Atlantic for SALTRACE-W, with $d_{eff}$ of 4.1 μm.




The stabilization of the size distribution is contrary to what would be expected from gravitational sedimentation theory. However, it *is* consistent with the findings of now numerous publications of individual field campaign dust size distributions, where larger particles were observed than could be explained by gravitational settling alone (Ryder et al., 2013a; Denjean et al., 2016; Weinzierl et al., 2017; Stevenson et al., 2015; Gasteiger et al., 2017; Ryder et al., 2018; van der Does et al., 2018; Maring et al., 2003). Ryder et al. (2013a) examined the mechanisms for transport between fresh, aged and SAL dust during Fennec-Sahara, and found that sedimentation and dispersion were able to account for the loss of the accumulation and giant mode changes observed between the Saharan boundary layer and the SAL during Fennec-Sahara, but not for the coarse mode which was retained to a greater degree than expected. Gasteiger et al. (2017) developed a simplified model for the long-range transport of Saharan dust aerosols over the Atlantic Ocean that was consistent with observations. Their results suggest that vertical mixing of the SAL air during the day (via convection caused by the absorption of sun light) was likely to be an important factor in explaining the dust measurements at different stages of the transport. van der Does et al. (2018) examined potential mechanisms for long-range transport of giant dust particles and found it would be most likely under highly optimal conditions incorporating high levels of turbulence and strong winds, which may also allow electrical levitation of dust particles. Long-range transport could be further enhanced by repeated lifting of dust particles by deep convective clouds. However, they stress that the details of these mechanisms are mostly unquantified and require further research.

Denjean et al. (2016) suggest that during ADRIMED high turbulent up and downdrafts of up to 5 cms$^{-1}$ (from model simulations) enabled large particle lifetime enhancement. During AER-D-SAL, measured vertical velocities within the SAL were over $\pm30$ cms$^{-1}$ in all cases, and sometimes up to $\pm80$ cms$^{-1}$. During Fennec-Sahara, vertical velocities were even larger: generally greater than 200 cms$^{-1}$ within the convective boundary layer (consistent with values from Marsham et al. (2013)), and frequently over 50 cms$^{-1}$ up to 5 km altitude. The gravitational settling velocity of a 10 µm diameter particle would be 1.1 cms$^{-1}$, and 28 cms$^{-1}$ for a 100 µm particle (Li and Osada, 2007). Therefore it appears possible that high levels of atmospheric turbulence could have sustained transport of larger particles for longer than expected by gravitational sedimentation. Additionally, during AER-D-SAL, vertical velocities were net positive in the SAL, supporting the possibility of solar absorption by the dust particles generating convection and daytime vertical mixing within the SAL (Gasteiger et al., 2017). The more absorbing nature of coarser particles in the solar spectrum would reinforce this mechanism.

## 4. Conclusions

Several airborne observational campaigns have recently revealed the ubiquitous nature of coarse and giant dust particles within dusty air masses. Here, we present mean PSDs and their uncertainties from one Saharan dataset and two SAL datasets where state-of-the art airborne measurements with consistent instrumentation were performed. These have been used to provide insights into how dust properties, and particularly the coarse and giant modes, change with transport and how this impacts optical properties.


We have contrasted the mean airborne ambient size distributions of dust measured over the Sahara during the Fennec fieldwork (both over the Sahara and in the SAL near the Canary Islands) to the more recent observations made during the AER-D fieldwork within the SAL. The observations utilize light shadowing techniques which allow measurement of giant mode dust

particles and avert some of the historical challenges of airborne measurements of dust. All datasets fully capture the coarse and giant dust particles, up to sizes of 100 µm (AER-D-SAL) and 300 µm (Fennec). As expected, Fennec-Sahara shows a greater giant mode (d>20 µm) than AER-D-SAL and Fennec-SAL, but the AER-D-SAL mean PSD shows a greater volume concentration at diameters smaller than 8 µm.

The vertical distribution of dust size shows that size distributions with an extremely strong giant mode ($d_{eff}$ between 12 to 21 µm) are only observed at low altitudes over the Sahara (up to around 1 km), and only for fresh events (under 12 h since uplift). However, for aged events (longer than 12 h since uplift), giant particles are still present in the PSD up to 5 km altitude with large $d_{eff}$ values of 5 to 10 µm. Effective diameters in AER-D-SAL were homogeneous at around 4 µm throughout the SAL. Models often use mass concentration as a diagnostic of aerosol amount, therefore we have provided these from observational

data in order to facilitate model validation studies. Mass concentration decreases with height over the Sahara, but is more homogeneous and well-mixed in the vertical in the SAL. Over the Sahara, 93% of dust mass is constituted by particles sized larger than 5 µm on average, and 40% of dust mass is constituted by particles sized larger than 20 µm. Since 5 µm and 20 µm are the diameters at which models begin to underestimate coarse mode concentrations and omit the giant mode respectively, models will be omitting a very large fraction of mass over the Sahara. During individual events, models may be missing up to

60% of mass by excluding dust sizes greater than 20 µm. Over the SAL, the fraction of mass omitted is smaller compared to the Sahara, but potentially still important: 61 to 89% of dust mass is constituted from sizes over 5 µm and 2 to 12% from sizes over 20 µm. Other processes, which were not examined directly here, such as the role of coarse and giant particles as ice nucleating particles, which affect the impact of dust on cloud development, will also be affected by model under-representation of coarse and giant dust particles.

The size-resolved contribution of the different PSDs to extinction coefficient has also been calculated. By excluding particles larger than 20 µm diameter, as in many dust models, 18% (8-23%) of extinction at a wavelength of 0.55 µm will be omitted over the Sahara and 1-4% (0-4%) will be omitted in the SAL. (Values in parentheses represent uncertainties due to both PSD variability and RI dataset). Similarly, for absorption at 0.55 µm, excluding the giant mode will omit 39% (18-48%) over the

Sahara and 2-10% (0-13%) over the SAL. In the longwave spectrum, at 10.8 µm, we find that only representing particles sized up to 20 µm diameter omits 26% (11-34%) of the extinction over the Sahara, and 2 to 6% (0-9%) of the extinction over the SAL.

The extinction coefficient profile determines the aerosol optical depth and the direct radiative effect of dust, while the absorption profile determines the semi-direct effect and impacts dust-driven shortwave atmospheric heating and may subsequently impact regional circulation (Perlwitz and Miller, 2010; Solmon et al., 2012; Woodage and Woodward, 2014). Our results suggest that the missing extinction and absorption in models will therefore alter the impact of dust in models.

Omitting the giant mode results in a greater omission of the longwave extinction radiative effects of dust than those of the shortwave. Additionally, in the shortwave, omission of absorption from the giant mode has most impact. Since both these processes lead to a warming of the earth-atmosphere system, this suggests that models are likely to be underestimating the warming influence of dust, estimated to be -0.1 (-0.3 to +0.1) Wm$^{-2}$ in the latest IPCC report (IPCC, 2013).

Additionally, these figures are lower bound estimates of the impact of neglected absorption and extinction in dust models, since they only account for giant particles being excluded, and not any additional underestimation of the coarse mode which is included, but poorly represented in models (e.g. Kok et al. (2017); (Evan et al., 2014)). Both excluding giant particles, or under representing the concentrations of coarse and giant particles, will lead to more important consequences over the Sahara compared to in the SAL.

This work makes the assumption that dust particles are spherical for the optical property calculations in order to enable multiple rapid computations. This assumption is likely to have little impact in the longwave spectrum, since the size parameter is smaller. In the shortwave, our results represent a lower bound for the impact of the coarser dust: Kok et al. (2017) show that non-spherical dust increases extinction efficiency by 50% for coarse particles. Additionally, most climate models still assume

spherical dust properties.

Finally, we put the Fennec-Sahara and AER-D-SAL PSDs in the context of other airborne campaigns of the last ten years which have measured Saharan dust, and included measurements larger than 10 µm diameter. The two sets of dust observations closest to dust sources, Fennec-Sahara and SAMUM1, show a clear presence of giant particles influencing the shape of the

PSDs, while those measuring transported dust showed a steeper drop off of the PSD and lower total concentrations. Despite this, there is still a significant presence of coarse and giant particles in the 'transported' size distributions. Evaluating effective diameter for each field campaign against dust age since uplift time reveals what appear to be two regimes of dust transport: firstly $d_{eff}$ drops off rapidly during initial transport within the first 36 hours, and secondly where $d_{eff}$ appears very stable despite significant amounts of transport between around 2 to 10 days.

It is clear that mineral dust coarse and giant modes are retained to a much greater degree than expected from gravitational sedimentation alone. The processes behind this are still unclear (e.g. van der Does et al. (2018)). Potential explanations which warrant further study include variations in fall speed dependent on particle composition, density, shape and orientation, turbulent and convective mixing, triboelectric charging (e.g. Harrison et al. (2018)), and radiative lofting impacts of the coarse

and giant particles. Similar processes and uncertainties also apply to atmospheric transport of volcanic ash, where similar unexplained long-range transport of coarse and giant particles have been observed (e.g. Stevenson et al. (2015); Beckett et al. (2015); Saxby et al. (2018)).

Overall, climate models generally do not incorporate dust particles sized over 20 µm. Historically this has been because of the assumption that larger particles are deposited rapidly. This work suggests that although particles larger than 20 µm do exist up to high altitudes even in transported dust, it is over the Sahara that the contribution of this size range to total mass, absorption and extinction are most significant. For transported dust in the SAL, the size distribution has evolved such that the giant particles contribute only a small amount to total extinction and dust mass concentration. However, models begin to

underestimate dust concentrations at sizes well below this, from 5 µm upwards. Our results show that dust particles in this size range (diameters 5 to 20 µm) are still highly prevalent, and contribute a large amount to extinction and dust mass in the SAL as well as over the Sahara, so that better representation of the coarse mode size distribution within dust models is also an area for improvement.

In the absence of other mechanisms and explanations, it is natural that to date climate models employ some form of gravitational settling for dry deposition of dust. However, other mechanisms must be occurring in the real world in order to transport coarse and giant particles as far and for as long as detected in observations. Therefore further work, ideally combining observations and modelling efforts, in order to explain this transport, is required.

## 5. Data Availability

We are in the process of uploading the campaign mean data presented here to the Centre for Environmental Data Analysis (CEDA). Flight-by-flight aircraft data is publicly available from https://catalogue.ceda.ac.uk/uuid/affe775e8d8890a4556aec5bc4e0b45c.

## 6. Author Contributions

CLR designed and carried out the analysis and wrote the manuscript. EJH discussed the methodology and results. SALTRACE
size distributions were provided by AW and BW. SALTRACE dust age estimates were provided by PS and AP. All authors read and commented on the manuscript.

## 7. Competing Interests

None



## 8. Acknowledgements

This research is funded by NERC Fellowship Grant NE/M018288/1. FLEXPART output was generated using ERA5 data (Copernicus Climate Change Service information [2018]), accessed through ECMWF's Meteorological Archival and Retrieval System (MARS). SALTRACE dust age estimates were calculated using Copernicus Atmosphere Monitoring Service information [2018]. PS and AP thank the Austrian Meteorological Service ZAMG for access to MARS. BW, AP, and AW were funded from the European Research Council (ERC) under the European Union's Horizon 2020 research and innovation framework programme under grant agreement No. 640458 (A-LIFE). The SALTRACE research flights were funded by the Helmholtz Association under Grant VH-NG-606 (Helmholtz-Hochschul-Nachwuchsforschergruppe AerCARE), and by DLR. The authors are grateful to M.Woodage for comments on the manuscript and J.Banks for discussions relating to longwave dust radiative interactions.

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



# Figures

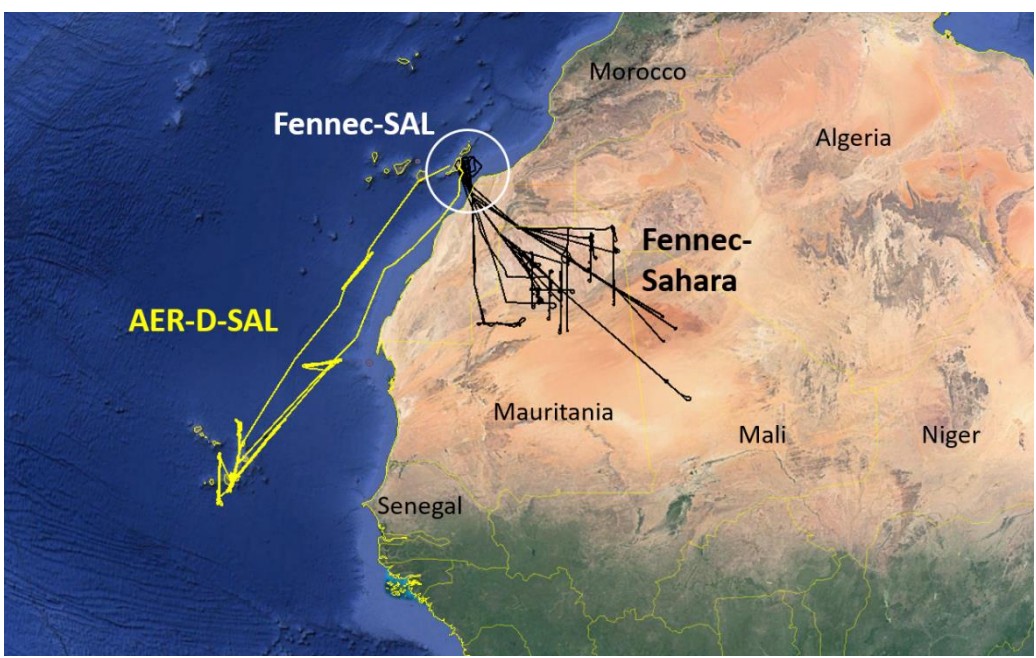

**Figure 1: Map showing locations of research flights: Fennec-Sahara in black, Fennec-SAL in black within white circle, AER-D SAL in yellow. Image provided using © Google Earth Pro, Map Data: © Google, SIO, NOAA, U.S. Navy, NGA, GEBCO, Landsat/COPERNICUS.**





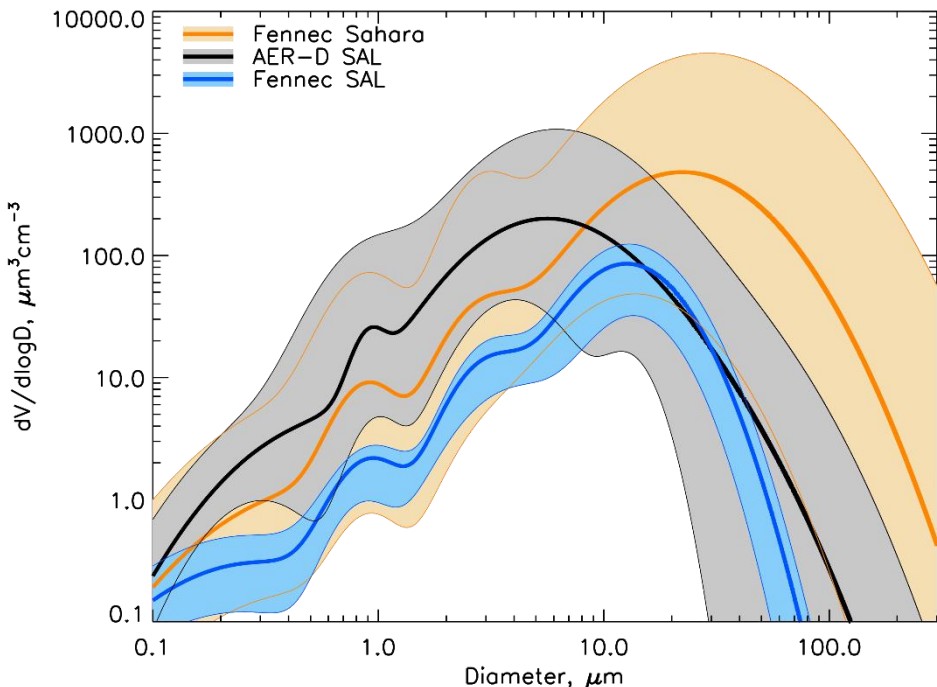

**Figure 2: Campaign ambient mean logfit size distributions for Fennec-Sahara (orange), AER-D SAL (black) and Fennec-SAL (blue). Bold lines indicate field campaign mean PSDs, shading indicates min:max range for SAL data and 10th percentile:maximum range for Fennec Sahara.**

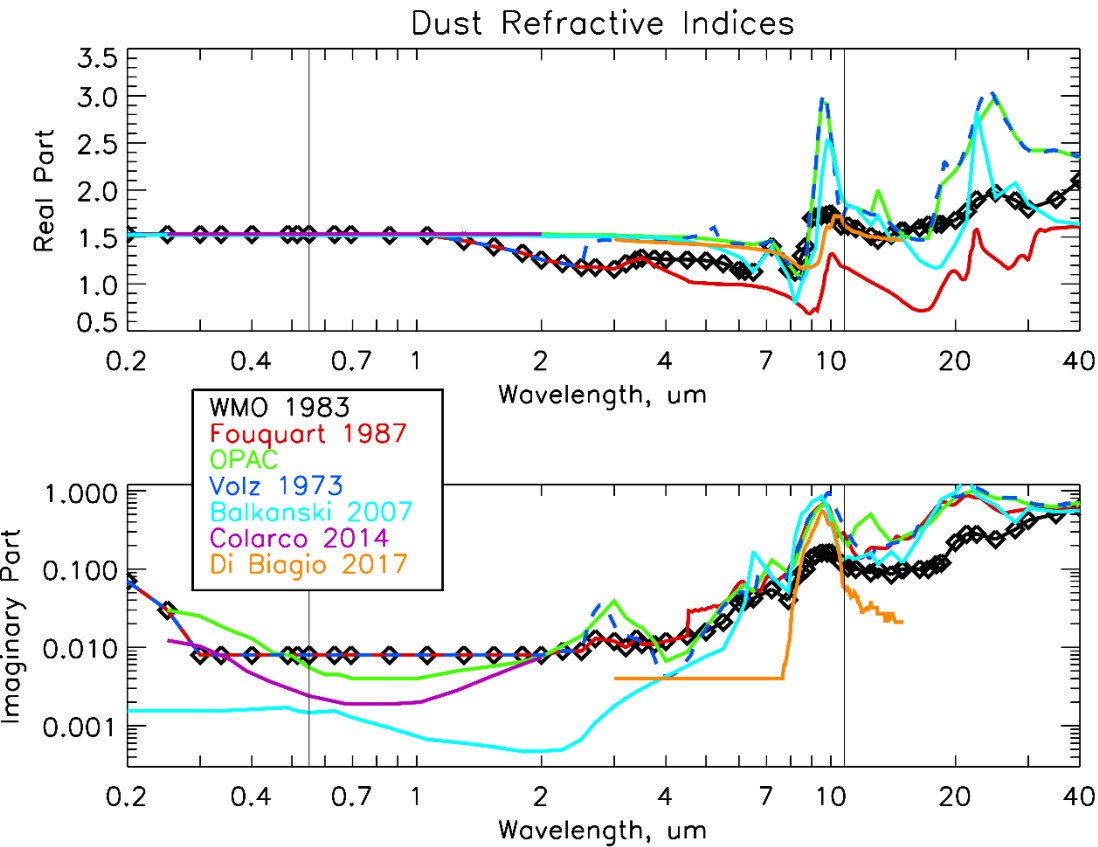

**Figure 3: Dust spectral refractive index datasets from the literature. Vertical lines indicate wavelengths of 0.55 and 10.8 μm. See text for dataset descriptions. Partial lines only provide a subset of spectral refractive indices.**





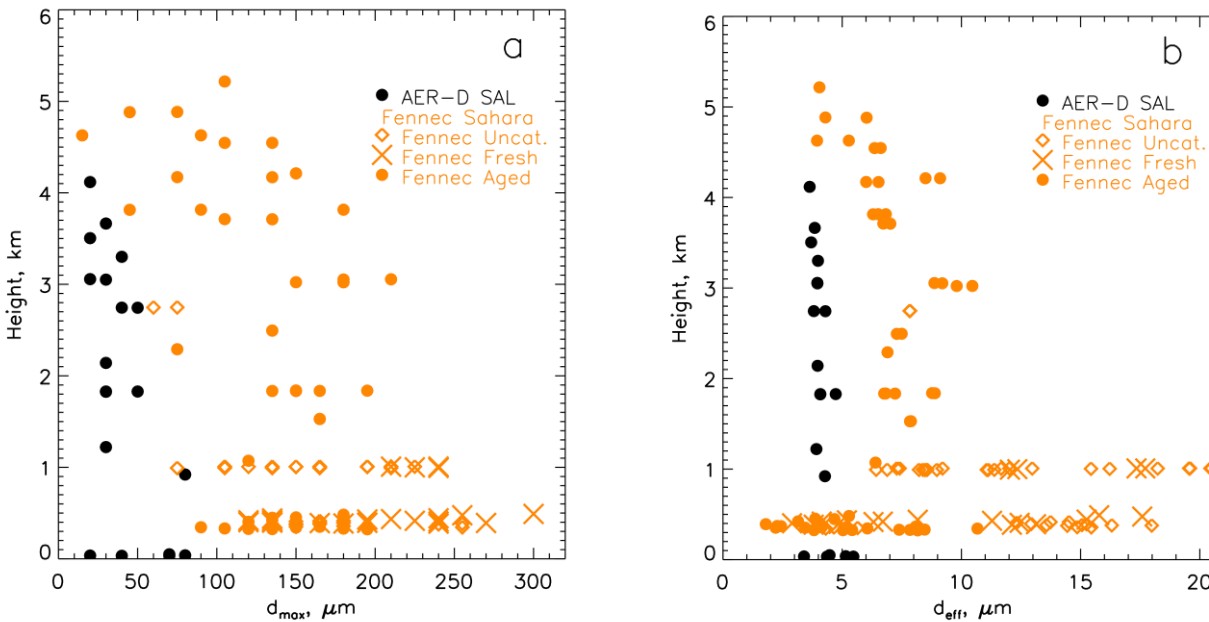

**Figure 4: Variation of dust size with altitude from Fennec-Sahara and AER-D-SAL, showing (a) maximum size detected ($d_{max}$) and (b) effective diameter ($d_{eff}$). $d_{eff}$ uncertainties are 5%, $d_{max}$ uncertainties are 10 µm for AER-D, 15 µm for Fenenc. Data are from horizontal flight legs.**





**Figure 5: Vertically resolved mass concentrations for Fennec-Sahara (orange), Fennec-SAL (blue) and AER-D-SAL. (black) (a) Total Mass concentration across all sizes measured; (b) accumulation mode mass concentration d<2.5 µm; (c) and (d) Fraction of mass at d>5 µm (c) and d>20 µm (d). Bold lines and shading indicate median and inter-quartile range respectively. Data is smoothed over 250m intervals, and for Fennec-Sahara only available down to 350m due to flight restrictions.**


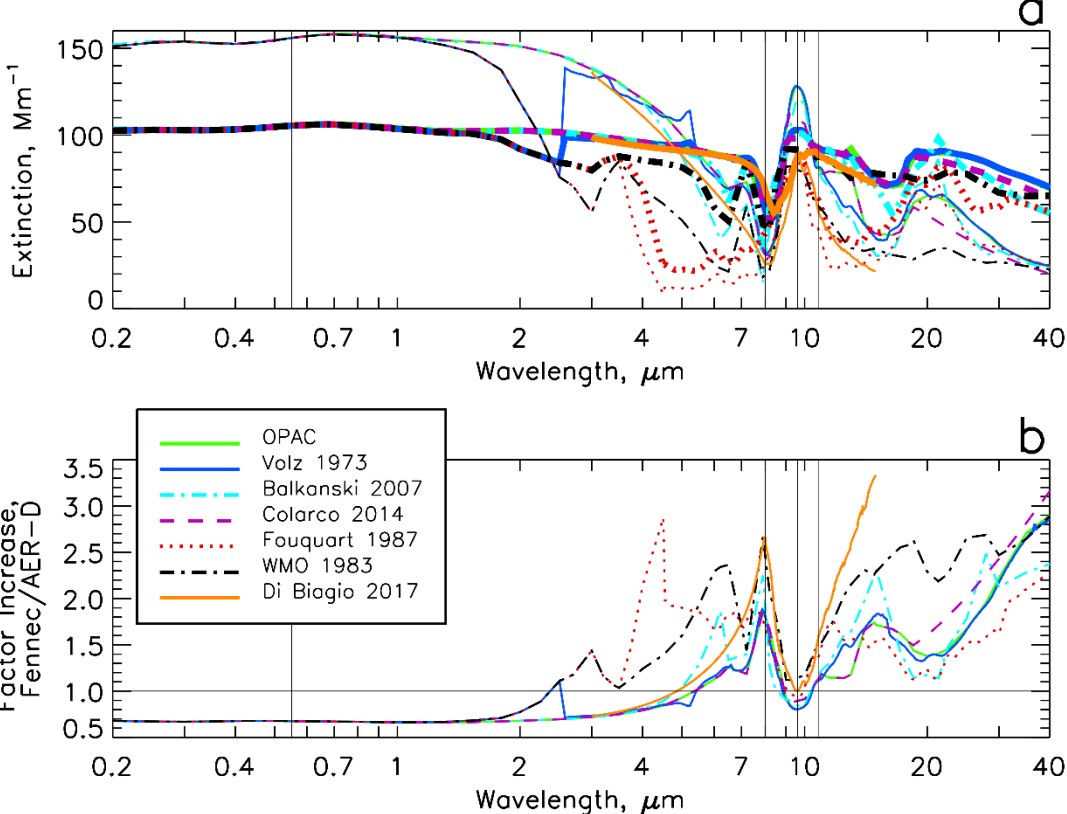

**Figure 6: Calculated Spectral Extinction coefficient, Mm⁻¹ (a) and factor increase in extinction (b) between Fennec-Sahara (bold lines) and AER-D-SAL (lightweight lines). Different colours indicate different RI datasets as in the legend. Vertical lines indicate 0.55, 8.0, 9.6, and 10.8 μm wavelengths.**



**Figure 7: Size resolved contribution to total absorption (thin lines) and extinction coefficient (bold lines) calculated for AER-D-SAL (black), Fennec-SAL (blue) and Fennec-Sahara (orange), at 0.55 μm, using the Colarco RI dataset. (a) Percentage contribution as a function of diameter, (b) cumulative percentage extinction coefficient as a function of diameter, (c) cumulative percentage absorption coefficient as a function of diameter. In (b) and (c), shading bounded by dashed lines shows the uncertainty due to the range of RI datasets and PSD variability observed in each observational campaign. Vertical lines indicate diameters of 2.5, 5, 10, 20 and 30 μm.**



**Figure 8: Size resolved contribution to total absorption (thin lines) and extinction coefficient (bold lines) calculated for AER-D-SAL (black), Fennec-SAL (blue) and Fennec-Sahara (orange), at 10.8 µm, using the Volz RI dataset. (a) Percentage contribution as a function of diameter, (b) cumulative percentage extinction as a function of diameter. In (b), shading bounded by dashed lines shows the uncertainty due to the range of RI datasets and PSD variability observed in each observational campaign. Vertical lines indicate diameters of 2.5, 5, 10, 20 and 30 µm.**

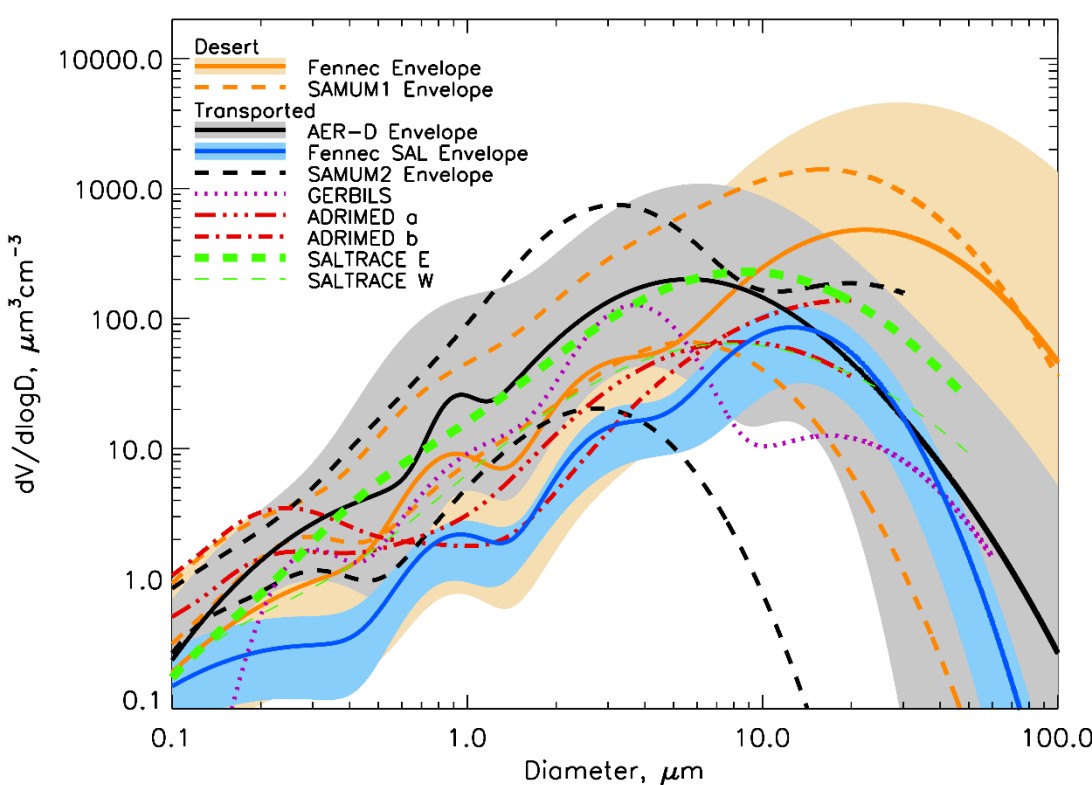

**Figure 9: Lognormal ambient volume size distributions for recent airborne campaigns measuring Saharan dust extending to sizes larger than 20µm diameter. Observations close to dust sources are coloured orange. AER-D SAL mean and minimum/maximum envelope is shaded grey, Fennec-Sahara 10th percentile/maximum envelope is shaded orange, Fennec-SAL minimum/maximum envelope is shaded blue as in Figure 2. ADRIMED a and b represent dust above 3km and beneath 3km respectively. SALTRACE E and W represent observations over the eastern vs western Atlantic. Lognormal curves are not shown at sizes above which measurements were made. See text for references for each campaign. SAMUM2 data are provided at standard temperature and pressure.**



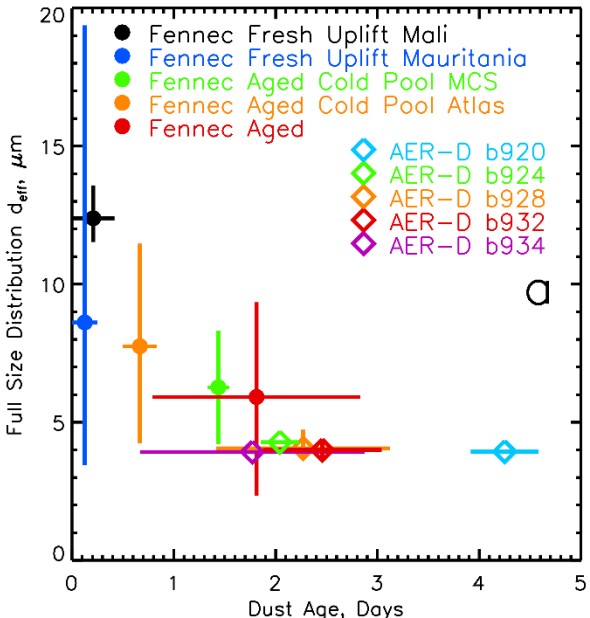
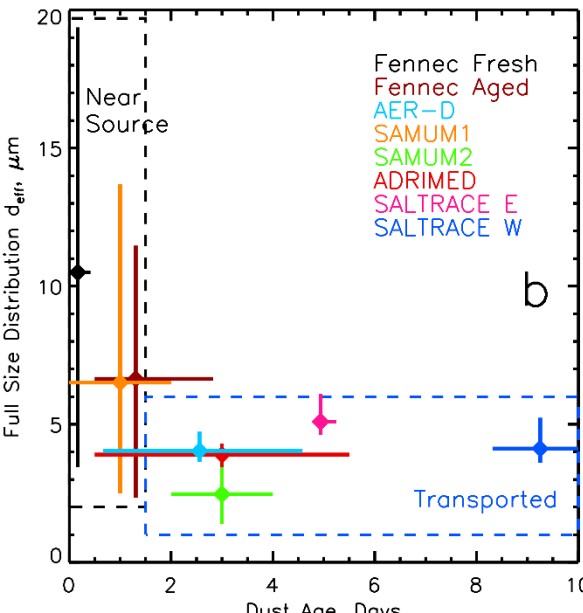

**Figure 10: Aircraft observations of effective diameter for the full size distribution against dust age since uplift. (a) Fennec and AER-D: Fennec is categorized by type of dust event (see Ryder et al., 2013b), AER-D data is separated by flight. (b) Saharan dust aircraft observations which fully measured coarse mode size distribution up to at least 20 μm diameter. $d_{eff}$ is shown for the full size distribution, or up to the maximum measurement diameter. Fennec-Sahara data are from Ryder et al. (2013b) and are identical to values shown in panel a, but with data merged into fresh and aged dust categories. AER-D-SAL data represent the range of flight-by-flight data shown in panel a. SAMUM1 data are from Weinzierl et al. (2009) Table 4. SAMUM2 data are from Weinzierl et al. (2011) Table 3. ADRIMED data are calculated from lognormal size distributions parameters in Denjean et al. (2016a) up to a maximum measurement size of 20 μm. SALTRACE (E and W: East and West) data are new calculations based on flight segments from Weinzierl et al. (2017). Data for panel b are given in supplement.**



# Tables

| Maximum (cut-off) Diameter, µm | % Contribution to Total SW Scattering | | | % Contribution to Total SW Absorption | | | % Contribution to Total SW Extinction | | |
|---|---|---|---|---|---|---|---|---|---|
| | Fennec-Sahara | AER-D SAL | Fennec-SAL | Fennec-Sahara | AER-D SAL | Fennec-SAL | Fennec-Sahara | AER-D SAL | Fennec-SAL |
| 2.5 | 31 (28,39) | 50 (46,52) | 34 (30,35) | 5 (2,6) | 20 (16,21) | 7 (4,8) | 27 (27,34) | 48 (44,50) | 31 (27,32) |
| 5 | 46 (43,52) | 80 (73,80) | 53 (47,55) | 12 (6,13) | 53 (40,56) | 20 (14,22) | 41 (41,45) | 78 (71,78) | 50 (413,51) |
| 10 | 63 (62,71) | 96 (95,97) | 77 (75,79) | 27 (23,43) | 85 (83,90) | 49 (41,52) | 58 (55,67) | 95 (95,97) | 74 (71,76) |
| 20 | 86 (82,94) | 100 (100,100) | 97 (96,97) | 61 (52,82) | 98 (97,100) | 90 (87,91) | 82 (77,92) | 99 (99,100) | 96 (96,97) |
| 30 | 95 (92,99) | 100 (100,100) | 100 (100,100) | 83 (74,96) | 99 (99,100) | 99 (98,99) | 93 (89,98) | 100 (100,100) | 100 (99,100) |
| 40 | 97 (95,100) | 100 (100,100) | 100 (100,100) | 90 (83,98) | 100 (100,100) | 100 (99,100) | 96 (93,99) | 100 (100,100) | 100 (100,100) |
| 60 | 99 (98,100) | 100 (100,100) | 100 (100,100) | 98 (94,100) | 100 (100,100) | 100 (100,100) | 99 (98,100) | 100 (100,100) | 100 (100,100) |

**Table 1: Percentage contribution to total shortwave scattering, absorption and extinction coefficient at 0.55 µm, as a function of maximum particle size considered, for the Fennec-Sahara, AER-D-SAL and Fennec-SAL mean size distributions using the Colarco et al. (2014) refractive index dataset. Values correspond to data shown in Figure 6. Uncertainties shown in parentheses represent lower and upper values due to uncertainties in PSD and RI dataset.**



| Maximum (cut-off) Diameter, μm | % Contribution to LW Scattering | | | % Contribution to LW Absorption | | | % Contribution to LW Extinction | | |
|---|---|---|---|---|---|---|---|---|---|
| | Fennec-Sahara | AER-D SAL | Fennec-SAL | Fennec-Sahara | AER-D SAL | Fennec-SAL | Fennec-Sahara | AER-D SAL | Fennec-SAL |
| 2.5 | 0 (0,0) | 2 (1,3) | 1 (0,1) | 4 (2,6) | 14 (9,19) | 5 (3,8) | 2 (1,4) | 9 (3,16) | 3 (1,6) |
| 5 | 6 (2,8) | 29 (18,43) | 10 (5,13) | 14 (5,15) | 49 (35,63) | 20 (10,25) | 10 (4,12) | 41 (22,56) | 15 (6,20) |
| 10 | 33 (15,50) | 8 2(66,86) | 52 (32,56) | 38 (20,53) | 87 (76,91) | 56 (40,57) | 35 (26,51) | 85 (74,89) | 54 (42,54) |
| 20 | 72 (55,89) | 98 (94,100) | 93 (86,95) | 75 (60,90) | 49 (47,50) | 94 (88,95) | 74 (66,89) | 98 (98,100) | 94 (91,94) |
| 30 | 89 (81,98) | 100 (99,100) | 99 (98,100) | 91 (84,98) | 87 (87,88) | 99 (99,100) | 90 (84,98) | 100 (100,100) | 99 (99,100) |
| 40 | 94 (89,99) | 100 (100,100) | 100 (100,100) | 95 (91,99) | 99 (98,99) | 100 (100,100) | 95 (90,99) | 100 (100,100) | 100 (100,100) |
| 60 | 99 (97,100) | 100(100,100) | 100 (100,100) | 99 (97,100) | 100 (100,100) | 100 (100,100) | 99 (97,100) | 100 (100,100) | 100 (100,100) |

**Table 2: Percentage contribution to total longwave scattering, absorption and extinction coefficient at 10.8 μm, as a function of maximum particle size considered, for the Fennec-Sahara, AER-D-SAL and Fennec-SAL mean size distributions using the Volz et al. (1973) refractive index dataset. Values correspond to data shown in Figure 7. Uncertainties shown in parentheses represent lower and upper values due to uncertainties in PSD and RI dataset.**