# Peer review of "Coarse and Giant Particles are Ubiquitous in Saharan Dust Export Regions and are Radiatively Significant over the Sahara"

_Atmospheric Chemistry and Physics, 2019_

## Referee Comment (RC1) · Anonymous Referee #2 · 12 Aug 2019

The authors address a topic of scientific significance. They present and analyze new data retrieved from experimental campaigns in the Sahara and the Saharan Air Layer, which provide information on the dust particle size distribution close to sources and in aged and transported dust masses. That information is relevant, among other aspects, to characterize the dust radiative effect, which remains nowadays uncertain.

The authors also apply a valid methodology, which is described in an appropriate way, and they put into context their results by considering previously published works.

Finally, the results are presented with a relevant number of figures and tables, as well as well as an appropriate use of English language. Some sections could be simplified

(e.g. the methods section) in order to make it more concise, but overall the article is well structured and clear.

For those reasons, I believe that the article fully meets the Atmospheric Chemistry and Physics quality criteria and merits being published. I would recommend some minor corrections, that could help improve further the manuscript quality. Please, find them below.

** General comments **

The authors identify the particle size distribution as one of the key factors in characterizing the dust radiative effect. However, there are other factors that influence the dust optical properties that could be further discussed in the introduction section.

In addition, they present a thorough review of complex refractive indexes applicable to dust from different sources. They discuss the variability of the dust optical properties in the short and long-wave considering the ranges of uncertainty of the PSD and RI together. In my view, they have the opportunity, with the data presented, to discuss further the contribution of each of those separately, providing a valuable insight for the modelling community.

Finally, I would recommend to comment further on the representativeness of the data presented when the authors introduce and describe the different campaigns.

** Specific Comments **

Introduction

Page 2; line 15: Jickells et al. (2005) focuses on oceanic ecosystems, rather than Amazon rainforest effects. The authors could provide additional references regarding the effect of dust deposition on the Amazon rainforest (as they do in line 24).

Page 3; line 28 to page 4; line 4: I would suggest to move this paragraph to page 3, line 2, and link it to the discussion on the uncertainties on optical properties due to the size

distribution. This will also allow to avoid repeating the "sensitivity of satellite retrievals to assumed PSD." It would be also advisable to acknowledge at some point in the introduction other sources of uncertainty in the dust optical properties (e.g. mineralogical composition, shape, mixing state).

Methods

The methods section includes all the relevant details to understand the measurements and analyses performed. However, I believe that it would be easier to follow if it could be simplified or slightly reorganized. I would suggest to:

Include a summary table with the most relevant details of the campaigns

Summarize all novel data and analyses in one paragraph if possible. For instance, page 5, line 32 explains a new metric from Fennec data, later on page 6, lines 6 to 9, new data and analyses are highlighted.

Rename or reorganize the sub-sections. Section "2.1 Size distribution" provides details about the spatial sampling (e.g. horizontal flight legs, vertical profiles, etc.), which, in my view, would be part of the fieldwork setup. The last paragraph of the same section mentions the optical properties calculations. I would move that information to section "2.2. Optical property calculations".

Results

Page 10; line 14: Health effects could be pointed here too, as they are highlighted later in lines 21-23.

Page 10; line 30: Would it be possible to provide a measure of the underestimation of particles above 5 $\mu$m in models?

Page 11; lines 5-10: Due to dust seasonality, a direct comparison of the DMP values obtained from the summertime campaigns and the modelled annual mean cannot be used to draw conclusions. Also, the authors refer to satellite data that is not mentioned

in the text. I would recommend to compare to seasonal (summertime) modelled values, if possible. In line with this comment, and as suggested in the General comments section, I would suggest to briefly comment on the representativeness of the data earlier, when the different campaigns are introduced.

Page 12; lines 8-12: I would suggest to specify that only information on panel a of Figure 7 relies exclusively on Colarco RI and the mean PSDs. Panels b and c, as the reader understands from line 10, include the uncertainty due to the variability of RIs and PSDs. Would it be possible to disentangle both sources of uncertainty? In my view, it would be very interesting to have a measure of the relative contribution to the uncertainty attributable to PSD and RI separately.

Page 13; line 10: Please, specify in the text, as done for Figure 7, the PSD and RI source used as a reference to calculate the size resolved contribution to optical properties at 10.8 $\mu$m.

Page 13; line 14: Please, specify the source of the range of SSA (0.4-0.5).

Page 13; lines 25-27: I would suggest to include also the information related to absorption in Figure 8. It would make it fully consistent with Figure 7. Alternatively, I would move the justification for not including this information to the paragraph presenting Figure 8 (i.e. lines 10 and below). As commented for the short-wave, it would be very interesting to distinguish in the uncertainty the relative contribution of the variability of PSDs and RIs.

Page 17; lines 23-24: Only the effect of coarse particles as ICN is mentioned. I would suggest to list other possible processes affected by a misrepresentation of coarse particles.

** Technical corrections **

Page 3; lines 20-25: I would suggest to identify the reference for each specific campaign, instead of listing all at the end of the paragraph.

Page 5; line 29: Ryder et al. (2018)

Page 6; lines 2-3: Ryder et al. (2018)

Page 6; line 24: Add the acronym for refractive index (RI) here, and remove it later in line 31.

Page 8; line 14: "The age [. . .] was" or "The ages [. . .] were"

Page 9; line 24: Specify what z refers to (z<100m).

Page 12; lines 24-25: The definition of panels b and c of Figure 7 has already been provided in lines 10-11 of the same page.

Page 13; line 27: Please, specify what does the 50% underestimate refer to.

Page 14; line 3: The parenthesis in "(and therefore do not [. . .]" should be removed or closed somewhere later.

Page 14; line 6: Ryder et al. (2018)

Page 17; lines 28 and 30-31: For the values: "1-4%(0-4%)" and "2-10%(0-13%)", please, specify in the text what do the ranges correspond to (mean values for the two SAL campaigns and range of variability due to RI and PSDs?).

Page 18; line 12: Please, include references in the same format. "Kok et al. (2014); (Evan et al., 2014)".

Figure 5 caption. Please, include a space between the number and units of 250m and 350m.

Figure 9 caption. Please, include a space between number and units of 3km and 20$\mu$m.

Please, find the comments to the article also in the attached pdf document.

Please also note the supplement to this comment:
https://www.atmos-chem-phys-discuss.net/acp-2019-421/acp-2019-421-RC1-

supplement.pdf

---

## Referee Comment (RC2) · Anonymous Referee #3 · 15 Aug 2019

This is overall an excellent paper that draws on previously published work to review the contribution of coarse dust to the dust loading and extinction in and near the Sahara. This paper will be a valuable addition to the literature.

The authors report some impressive findings of the contribution of coarse and giant particles to mass loading and extinction, particularly over the Sahara. These particles seem to account for much more of the dust loading and SW and LW extinction than realized or accounted for in models, so this is important. But if I'm not mistaken, all the observations used were taken during the summer months. Because convection is stronger in those months, dust layers are higher, and coarse dust can be expected

to be a larger fraction of the dust loading than in winter months. This is for instance shown explicitly by surface observations in Van der Does et al. So it's important that the authors emphasize either that their findings apply to the summer months, and/or that their findings would be an upper limit for the annually-averaged contribution of coarse dust. Currently, that's not clear.

Further comments:

- The abstract is clear but very long ($\sim$400 words), so I'd recommend shortening to make the main findings easier to absorb.

- The D_max metric is defined as the largest bin for which >4 particles were detected during a flight leg. This seems a bit problematic as it depends strongly on instrument sensitivity and flight duration. This makes it difficult to interpret and also difficult to compare between different observations with different flight durations or instruments, which the authors acknowledge on p. 10. Perhaps a metric like the 99th percentile of the cumulative mass distribution would be more meaningful and useful?

- Similar to many previous studies, the authors assume that dust is spherical for calculations of optical properties. That's reasonable, but considering that dust is quite aspherical, they should include a few sentences on how they expect their results to change if they had accounted for dust asphericity.

- Line 4, p.2: There's a wide range of estimates of annual dust emissions, so 1,100 Tg/year is too precise a number. More importantly, the dust size range to which this number applies should be included, especially considering the topic of the article.

- Line 7, p. 18: The authors here seem to confuse radiative forcing and radiative effect. See for instance Heald et al. (2014). The authors seem to allude here to the dust radiative effect, which is the net effect on the climate of dust interactions with radiation. The IPCC report calculated the radiative forcing, which is the change in that radiative effect. Please correct accordingly.

- Figure 4: It's not clear to me why this figure does not include results from FENNEC SAL?

- Figure 7: The vertical axis "% contribution" is only meaningful if the spacing of each bin is provided. I recommend changing this axis to something meaningful like "% contribution per ln D (or dQ/dlnD)". Same comment for Fig. 9. Also, I'd suggest adding the titles "Extinction" and "Absorption" to panels b and c.

---

## Referee Comment (RC3) · Anonymous Referee #1 · 18 Aug 2019

Overall, the manuscript provides significant information and makes a valuable contribution to desert dust research. Ryder et al. reveal the radiative effect of the "forgotten" coarse dust mode that is not taken into account either in remote sensing retrievals or global models, as it concerns its specific impact on the extinction (and consequently on radiation). I believe that the paper is ready for publication and I provide at the following paragraphs only my suggestions for its improvement:

One limitation of the study concerns the methodology followed to retrieve aerosol extinction from the measured size distributions. Mie scattering codes are inadequate for this type of extinction simulations, due to the fact that desert dust is non-spherical by

its nature at all particle modes. The impact of non-sphericity on extinction might not be that high in shortwave, however this statement has not been proven yet using realistic particle shapes, it is only a feeling that the community has at the moment since there are no scattering simulations for non-spherical particles that cover all sizes and specrum (this requires a vast amount of computing resources for processing all particle sizes). However, and as the authors mention already, calculations for spherical particles are still of high importance for the scientific community, since these results are comparable to global model simulations, where non-sphericity is not taken into account as well. I suggest though that the authors would add some sentences on the need to further study the impacts of non-sphericity based on more realistic representations of mineral particle shapes (e.g., to add information on the related paragraph such as if the authors intent to run such a study in the future, if yes, is there any information on non-sphericity from the campaigns mentioned in the manuscript etc).

One second suggestion is to use the lidar extinction retrievals from FAAM. The FAAM lidar is a backscatter system at 355 nm but for Saharan dust we are well aware of the lidar ratio so as to estimate an extinction profile from the backscatter retrievals. This is a valuable information for the shortwave range, since we all consider that dust extinction and backscatter have negligible spectral dependence in this spectral range. The FAAM lidar profiles can add an extinction closure in this beautiful work so as to increase its reliability.

---

## Author Comment (AC1) · 23 Sep 2019

**Response to Reviewers**

We would like to thank the reviewers for their very positive comments and careful reading of the manuscript, and extremely useful suggestions. Below, we respond to each of the reviewers' comments in red.

**RC1 - Anonymous Referee #2**

The authors address a topic of scientific significance. They present and analyze new data retrieved from experimental campaigns in the Sahara and the Saharan Air Layer, which provide information on the dust particle size distribution close to sources and in aged and transported dust masses. That information is relevant, among other aspects, to characterize the dust radiative effect, which remains nowadays uncertain. The authors also apply a valid methodology, which is described in an appropriate way, and they put into context their results by considering previously published works.

Finally, the results are presented with a relevant number of figures and tables, as well as well as an appropriate use of English language. Some sections could be simplified (e.g. the methods section) in order to make it more concise, but overall the article is well structured and clear.

For those reasons, I believe that the article fully meets the Atmospheric Chemistry and Physics quality criteria and merits being published. I would recommend some minor corrections, that could help improve further the manuscript quality. Please, find them below.

We thank the reviewer for their positive and useful comments. We respond to each of their points in turn below.

** General comments **

The authors identify the particle size distribution as one of the key factors in characterizing the dust radiative effect. However, there are other factors that influence the dust optical properties that could be further discussed in the introduction section. In addition, they present a thorough review of complex refractive indexes applicable to dust from different sources. They discuss the variability of the dust optical properties in the short and long-wave considering the ranges of uncertainty of the PSD and RI together. In my view, they have the opportunity, with the data presented, to discuss further the contribution of each of those separately, providing a valuable insight for the modelling community.

Finally, I would recommend to comment further on the representativeness of the data presented when the authors introduce and describe the different campaigns.

Each of the above points are dealt with below when mentioned in the specific comments.

** Specific Comments **

Introduction

Page 2; line 15: Jickells et al. (2005) focuses on oceanic ecosystems, rather than Amazon rainforest effects. The authors could provide additional references regarding the effect of dust deposition on the Amazon rainforest (as they do in line 24).

Done

Page 3; line 28 to page 4; line 4: I would suggest to move this paragraph to page 3, line 2, and link it to the discussion on the uncertainties on distribution. This will also allow to avoid repeating the "sensitivity of satellite retrievals to assumed PSD."

This paragraph has been moved as suggested, and the first instance of 'sensitivity of satellite retrievals…' has been deleted.

It would be also advisable to acknowledge at some point in the introduction other sources of uncertainty in the dust optical properties (e.g. mineralogical composition, shape, mixing state).

We now include the text, "Dust optical properties are influenced by several factors, including chemical composition, mixing state, particle shape and size" At the beginning of the third introduction paragraph.

Methods

The methods section includes all the relevant details to understand the measurements and analyses performed. However, I believe that it would be easier to follow if it could be simplified or slightly reorganized. I would suggest to: Include a summary table with the most relevant details of the campaigns

We have now included a new table (now Table 1), summarizing the relevant campaigns and details. This is reproduced from Ryder et al. (2018) Table 1, but with Fennec-SAL and Fennec-Sahara now separated. References to the various campaigns throughout the paper are now only provided as acronyms and generally without references, which are in Table 1.

Summarize all novel data and analyses in one paragraph if possible. For instance, page 5, line 32 explains a new metric from Fennec data, later on page 6, lines 6 to 9, new data and analyses are highlighted.

The sentence on p5 has been removed, and the paragraph on p6 reworded and moved to the end of section 2.1 to make the new data used much clearer. This paragraph now reads, "This article expands on the existing published work and data from Fennec and AER-D. Our emphasis is on using the combination of data in the context of transport time and vertical distribution. New data specifically includes: the Fennec-SAL lognormal mean PSD and uncertainties, vertical distributions of $d_{max}$ for Fennec-Sahara, vertical distributions of $d_{eff}$ for Fennec-Sahara separated by fresh and aged dust events, vertical distributions of mass concentration and DMP for Fennec-Sahara and Fennec-SAL."

Rename or reorganize the sub-sections. Section "2.1 Size distribution" provides details about the spatial sampling (e.g. horizontal flight legs, vertical profiles, etc.), which, in my view, would be part of the fieldwork setup. The last paragraph of the same section mentions the optical properties calculations. I would move that information to section 2.2. Optical property calculations".

Section 2.1 has been renamed 'Size Distribution Measurements' to make this clearer. We have removed the reference to optical properties at the end of section 2.1.

Results

Page 10; line 14: Health effects could be pointed here too, as they are highlighted later in lines 21-23.

Done

Page 10; line 30: Would it be possible to provide a measure of the underestimation of particles above 5 m in models?

Kok et al. (2017) present differences between AeroCom models and an experimentally constrained PSD containing models. In this case, at 5µm diameter the models underestimate dV/dlnD by up to

around a factor of 5. Above this diameter, there is around an order of magnitude difference. This has been added to the text.

Page 11; lines 5-10: Due to dust seasonality, a direct comparison of the DMP values obtained from the summertime campaigns and the modelled annual mean cannot be used to draw conclusions. Also, the authors refer to satellite data that is not mentioned optical properties due to the size in the text. I would recommend to compare to seasonal (summertime) modelled values, if possible. In line with this comment, and as suggested in the General comments section, I would suggest to briefly comment on the representativeness of the data earlier, when the different campaigns are introduced.

Getting hold of summertime-only DMPs for the Eastern Atlantic only is challenging – most DMPs are typically reported as annual and/or global averages, and when broken down to regional, temporal values, are typically converted to AODs in publications in order to compare to available observations. Nevertheless, we are grateful to able to access some seasonal unpublished DMP data from Amato Evan for the region. We added the sentence, "Unpublished analysis of summertime-only DMPs from a subset of CMIP5 models suggest values higher by around 35% (personal communication, A. Evan)- not nearly enough to reconcile the observational-model differences."

The following sentence has also been added to Section 2.1, "Although each campaign lasted only around 3 weeks, the data captured by each has been shown to be climatologically representative (Ryder et al., 2015; Ryder et al., 2018)."

Page 12; lines 8-12: I would suggest to specify that only information on panel a of Figure 7 relies exclusively on Colarco RI and the mean PSDs. Panels b and c, as the reader understands from line 10, include the uncertainty due to the variability of RIs and PSDs.

The final sentence of this paragraph now reads, "Panel a uses the Colarco RI exclusively, while in panels b and c, the shading represents the uncertainty to both the ranges of PSD shown in Figure 2 and the range of refractive indices tested."

Would it be possible to disentangle both sources of uncertainty? In my view, it would be very interesting to have a measure of the relative contribution to the uncertainty attributable to PSD and RI separately.

Figure 6 provides our best description of the relative uncertainties in extinction due to PSD and RI, as described in Section 3.2.1, where we describe how the PSD uncertainty dominates in the shortwave spectrum, while both are important in the longwave spectrum. We agree that the relative uncertainty from RI vs PSD is an important question, and have added corresponding plots for absorption (in addition to extinction) to Figure 6, as well as an extra paragraph in section 3.2.1 to expand on this.

We have also looked at the relative uncertainties for the size-resolved optical properties. At 0.55 μm, for extinction the size-resolved uncertainty is almost totally due to PSD uncertainty, while the absorption size-resolved uncertainty varies with cut-off diameter and with campaign, being dominated by RI uncertainty at d<2.5 μm for AER-D-SAL and Fennec-SAL and at d<5 μm for Fennec-Sahara. Above these diameters PSD uncertainty dominates, contributing up to twice the uncertainty from RI. At 10.8 μm, for extinction the PSD and RI uncertainty contribute roughly equally to the total size-resolved uncertainty, though this varies with cut-off diameter. However, we do not consider the relative uncertainties in this size-resolved percentage contribution context to be informative – rather the relative uncertainties are most important to the absolute optical properties and at all spectral wavelengths, as now given in Figure 6 and Section 3.2.1. Therefore we simply extend the discussion of the relative uncertainties in that section along with the addition of the spectral absorption plots.

Page 13; line 10: Please, specify in the text, as done for Figure 7, the PSD and RI source used as a reference to calculate the size resolved contribution to optical properties at 10.8 m.

This sentence has been added, "As in Figure 7, the three campaign mean PSDs have been used (from Figure 2) with the Colarco RI. Panel a uses the Colarco RI exclusively, while in panel b the shading represents the uncertainty to both the ranges of PSD shown in Figure 2 and the different RI datasets."

Page 13; line 14: Please, specify the source of the range of SSA (0.4-0.5).

These values come from data which goes into Figure 8 – this has been changed to "giving SSA values…"

Page 13; lines 25-27: I would suggest to include also the information related to absorption in Figure 8. It would make it fully consistent with Figure 7. Alternatively, I would move the justification for not including this information to the paragraph presenting Figure 8 (i.e. lines 10 and below).

We have changed Figure 8 now to include panel c, showing absorption.

As commented for the short-wave, it would be very interesting to distinguish in the uncertainty the relative contribution of the variability of PSDs and RIs.

See above comment relating to the SW component of uncertainties.

Page 17; lines 23-24: Only the effect of coarse particles as ICN is mentioned. I would suggest to list other possible processes affected by a misrepresentation of coarse particles.

We have added the role of dust as cloud condensation nuclei to this sentence, as well as a sentence relating to biogeochemical cycles and human health.

** Technical corrections **

These have all been changed, and are only commented on individually below where necessary.

Page 3; lines 20-25: I would suggest to identify the reference for each specific campaign, instead of listing all at the end of the paragraph.

As described above, we have now included a new table (now Table 1), summarizing the relevant campaigns and details. This is reproduced from Ryder et al. (2018) Table 1, but with Fennec-SAL and Fennec-Sahara now separated. References to the various campaigns throughout the paper are now only provided as acronyms and generally without references, which are in Table 1.

Page 5; line 29: Ryder et al. (2018)

Page 6; lines 2-3: Ryder et al. (2018)

Page 6; line 24: Add the acronym for refractive index (RI) here, and remove it later in line 31.

Page 8; line 14: "The age [: : :] was" or "The ages [: : :] were"

Page 9; line 24: Specify what z refers to (z<100m).

Page 12; lines 24-25: The definition of panels b and c of Figure 7 has already been provided in lines 10-11 of the same page.

Page 13; line 27: Please, specify what does the 50% underestimate refer to.

Now included – it refers to dust radiative effect.

Page 14; line 3: The parenthesis in "(and therefore do not [: : :]" should be removed or closed somewhere later.

Page 14; line 6: Ryder et al. (2018)

Page 17; lines 28 and 30-31: For the values: "1-4%(0-4%)" and "2-10%(0-13%)", please, specify in the text what do the ranges correspond to (mean values for the two SAL campaigns and range of variability due to RI and PSDs?).

Yes, this is correct. This has been reworded to, "Ranges correspond to mean values for both SAL campaigns, and values in parentheses represent the range of uncertainty due to both PSD variability and RI dataset."

Page 18; line 12: Please, include references in the same format. "Kok et al. (2014); (Evan et al., 2014)".

Figure 5 caption. Please, include a space between the number and units of 250m and 350m.

Figure 9 caption. Please, include a space between number and units of 3km and 20m.

**RC2 - Anonymous Referee #3**

This is overall an excellent paper that draws on previously published work to review the contribution of coarse dust to the dust loading and extinction in and near the Sahara. This paper will be a valuable addition to the literature. The authors report some impressive findings of the contribution of coarse and giant particles to mass loading and extinction, particularly over the Sahara. These particles seem to account for much more of the dust loading and SW and LW extinction than realized or accounted for in models, so this is important.

We thank the reviewer for their positive and useful comments. We have dealt with each comment in turn below.

But if I'm not mistaken, all the observations used were taken during the summer months. Because convection is stronger in those months, dust layers are higher, and coarse dust can be expected to be a larger fraction of the dust loading than in winter months. This is for instance shown explicitly by surface observations in Van der Does et al. So it's important that the authors emphasize either that their findings apply to the summer months, and/or that their findings would be an upper limit for the annually-averaged contribution of coarse dust. Currently, that's not clear.

The reviewer is correct and we agree with this point. To emphasize this more clearly, we have added the following paragraphs to the conclusion, with the second paragraph noting how this upper limit contrasts with the results being a lower limit due to uncertainties stemming from non-sphericity assumptions and not including any underestimation of the coarse mode.

"Another important factor for consideration is that the Fennec and AER-D observations are taken in summertime when Saharan and SAL dust loadings are at a maximum, and coarse and giant particles are also present in a greater fraction, due to strong convection lifting dust up to high altitudes over the Sahara, enabling further transport of the larger dust particles (e.g. McConnell et al. (2008); van der Does et al. (2016)). This is also reflected in the slightly lower sizes seen in SAMUM2 during winter. Therefore the impact of coarse and giant dust particles on mass concentrations and radiative effects presented here should be viewed as an upper bound within the seasonal cycle of dust.

Overall the three main uncertainties impacting this work are the exclusion of any underestimation of the coarse mode by models, a spherical assumption for scattering calculations, and the use of data based on summertime dust transport. The former two mean that our results of the impact of coarse and giant dust particles are underestimates, while the latter means our results are overestimates compared to an annual average."

Further comments:

- The abstract is clear but very long (400 words), so I'd recommend shortening to make the main findings easier to absorb.

We have shortened the abstract – please see uploaded manuscript.

- The D_max metric is defined as the largest bin for which >4 particles were detected during a flight leg. This seems a bit problematic as it depends strongly on instrument sensitivity and flight duration. This makes it difficult to interpret and also difficult to compare between different observations with different flight durations or instruments, which the authors acknowledge on p. 10. Perhaps a metric like the 99th percentile of the cumulative mass distribution would be more meaningful and useful?

We agree that the $d_{max}$ metric incurs some difficulties, as discussed. However, to be consistent with previous publications on these field campaigns, we prefer to remain using $d_{max}$, while ensuring its limitations are clearly explained, as already done in the article.

- Similar to many previous studies, the authors assume that dust is spherical for calculations of optical properties. That's reasonable, but considering that dust is quite aspherical, they should include a few sentences on how they expect their results to change if they had accounted for dust asphericity.

We refer the reviewer to the following paragraph in the conclusion:

"This work makes the assumption that dust particles are spherical for the optical property calculations in order to enable multiple rapid computations. This assumption is likely to have little impact in the longwave spectrum, since the size parameter is smaller. In the shortwave, our results represent a lower bound for the impact of the coarser dust: Kok et al. (2017) show that non-spherical dust increases extinction efficiency by 50% for coarse particles. Additionally, most climate models still assume spherical dust properties."

- Line 4, p.2: There's a wide range of estimates of annual dust emissions, so 1,100 Tg/year is too precise a number. More importantly, the dust size range to which this number applies should be included, especially considering the topic of the article.

This figure has been revised to a range of 1,000 to 4,000 Tg/year. We prefer not to introduce a size range here since we are simply introducing the dust cycle at this early stage in the paper. The implications of dust cut-off size on mass concentration are an important part of the results of the paper and are considered in detail throughout the article.

- Line 7, p. 18: The authors here seem to confuse radiative forcing and radiative effect. See for instance Heald et al. (2014). The authors seem to allude here to the dust radiative effect, which is the net effect on the climate of dust interactions with radiation. The IPCC report calculated the radiative forcing, which is the change in that radiative effect. Please correct accordingly.

The impacts of coarse and giant dust on radiation impact both the radiative effect and radiative forcing. This sentence has now been clarified – the first instance of 'radiative effect' is removed, and the IPCC statement clarified:

"Omitting the giant mode results in a greater omission of the longwave extinction than of the shortwave. … Since both these processes lead to a warming of the earth-atmosphere system, this suggests that models are likely to be underestimating the warming influence of dust, with the radiative forcing due to aerosol (dust)-radiation interactions estimated to be -0.1 (-0.3 to +0.1) Wm-2 in the latest IPCC report (IPCC, 2013)."

- Figure 4: It's not clear to me why this figure does not include results from FENNEC SAL?

Since Figure 4 shows data from horizontal flight legs, data is not shown for Fennec-SAL where only profiles were performed. This has been added to the caption.

- Figure 7: The vertical axis "% contribution" is only meaningful if the spacing of each bin is provided. I recommend changing this axis to something meaningful like "% contribution per ln D (or dQ/dlnD)". Same comment for Fig. 9. Also, I'd suggest adding the titles "Extinction" and "Absorption" to panels b and c.

'Extinction' and 'Absorption' titles have been added to panels b and c in Figures 7 and 8. In figures 7 and 8, the bin size intervals are small enough such that the resulting data forms a smooth curve, as shown. Data is not given in 1/lnD so adding this would be inaccurate. Figure 9 already states 'dV/dlogD' on the y-axis and data is provided as such.

**RC3 - Anonymous Referee #1**

Overall, the manuscript provides significant information and makes a valuable contribution to desert dust research. Ryder et al. reveal the radiative effect of the "forgotten" coarse dust mode that is not taken into account either in remote sensing retrievals or global models, as it concerns its specific impact on the extinction (and consequently on radiation). I believe that the paper is ready for publication and I provide at the following paragraphs only my suggestions for its improvement:

We thank the reviewer for these positive comments and are pleased they consider the manuscript is ready for publication.

One limitation of the study concerns the methodology followed to retrieve aerosol extinction from the measured size distributions. Mie scattering codes are inadequate for this type of extinction simulations, due to the fact that desert dust is non-spherical by its nature at all particle modes. The impact of non-sphericity on extinction might not be that high in shortwave, however this statement has not been proven yet using realistic particle shapes, it is only a feeling that the community has at the moment since there are no scattering simulations for non-spherical particles that cover all sizes and specrum (this requires a vast amount of computing resources for processing all particle sizes). However, and as the authors mention already, calculations for spherical particles are still of high importance for the scientific community, since these results are comparable to global model simulations, where non-sphericity is not taken into account as well. I suggest though that the authors would add some sentences on the need to further study the impacts of non-sphericity based on more realistic representations of mineral particle shapes (e.g., to add information on the related paragraph such as if the authors intent to run such a study in the future, if yes, is there any information on non-sphericity from the campaigns mentioned in the manuscript etc).

We agree with the reviewer, and have added the following sentences to the paragraph on non-sphericity in the conclusion:

"Measuring aspect-ratio across the full size range from in-situ measurements remains a challenging process. For the field campaigns studied here, aspect ratios were available only for a few samples from

AER-D (Ryder et al., 2018) and future work will consider dust shape during Fennec. We emphasize the need for further work to obtain observations of dust particle shape, particularly across the full size range of dust as presented here, and in calculating the optical properties for non-spherical dust across all size and spectral ranges, which requires extensive computing resources."

One second suggestion is to use the lidar extinction retrievals from FAAM. The FAAM lidar is a backscatter system at 355 nm but for Saharan dust we are well aware of the lidar ratio so as to estimate an extinction profile from the backscatter retrievals. This is a valuable information for the shortwave range, since we all consider that dust extinction and backscatter have negligible spectral dependence in this spectral range. The FAAM lidar profiles can add an extinction closure in this beautiful work so as to increase its reliability.

We agree that the FAAM lidar adds invaluable information to dust (indeed, all aerosol and cloud) observations during airborne campaigns. Both in-situ and lidar observations are presented for AER-D in Marenco et al. (2018), and for several Fennec publications (see Ryder et al., 2015). However, here we focus on the compilation of multiple data from three different campaigns rather than on the specific radiative closure between in-situ observations and lidar. Extension of this work to include the lidar observations would be beyond the scope of this article. Undoubtedly, the lidar data will be useful in future work such as radiative closure studies.